

**INVENTORY OF AFRICAN DESERT DUST EVENTS IN THE NORTH-CENTRAL IBERIAN**
**PENINSULA IN 2003-2014 BASED ON SUNPHOTOMETER-AERONET AND PARTICULATE**
**MASS-EMEP DATA**
**V.E. Cachorro[1*], M.A. Burgos[1], D. Mateos[1], C. Toledano[1], Y. Bennouna[1], B. Torres[1], A.M. de Frutos[1]**
**and A. Herguedas[2]**
[1]Grupo de Óptica Atmosférica, Facultad de Ciencias, Universidad de Valladolid, Paseo Belén 7, CP 47011,
Valladolid, Spain.
[2]Departamento de Control de Calidad y Cambio Climático. Consejería de Fomento y Medio Ambiente de la Junta de
Castilla y León. Valladolid, Spain.
*Correspondence to: Victoria E. Cachorro Revilla (chiqui@goa.uva.es)
**Abstract**
A reliable identification of Desert Dust (DD) episodes over North-central Spain is carried out based on
AErosol RObotic NETwork (AERONET) columnar aerosol sun-photometer (aerosol optical depth, AOD,
and Ångström exponent, α) and European Monitoring and Evaluation Programme (EMEP) surface
particulate mass concentration (PMx, x=10, 2.5, and 2.5-10 µm) as main core data. The impact of DD on
background aerosol conditions is detectable by means of aerosol load thresholds and complementary
information provided by HYSPLIT (Hybrid Single Particle Lagrangian Integrated Trajectory Model) air
mass back-trajectories, MODIS (Moderate Resolution Imaging Spectroradiometer) images, forecasting
aerosol models, and synoptic maps, which had been carefully reviewed by a human observer for each day
included in the DD inventory. This identification method allows the detection of low and moderate DD
intrusions and also mixtures of mineral dust with other aerosol types by means of the analysis of α. During
the period studied (2003-2014), a total of 152 DD episodes composed of 419 days are identified. Overall,
this means ~13 episodes and ~35 days per year with DD intrusion, representing 9.6% days/year. During the
identified DD outbreaks, 19 daily exceedances over 50 µg m$^{-3}$ are reported at the surface. The occurrence
of DD event days along the year peaks in March and June with a marked minimum in April and lowest
occurrence in winter. A large inter-annual variability is observed showing a statistically significant
temporal decreasing trend of ~3 days/year. As a key point, the DD impact on the aerosol climatology is
addressed by evaluating the DD contribution to AOD, PM$_{10}$, PM$_{2.5}$, and PM$_{2.5-10}$ obtaining mean values of
0.015 (11.5%), 1.3 µg m$^{-3}$ (11.8%), 0.55 µg m$^{-3}$ (8.5%) and 0.79 µg m$^{-3}$ (16.1%), respectively. Almost
similar annual cycles of DD contribution are obtained for AOD and PM$_{10}$ with two maxima, one in





summer (0.03 and 2.4 µg m$^{-3}$ for AOD in June and PM$_{10}$ in August, respectively) and another in March
(0.02 for AOD and 2.2 µg m$^{-3}$ for PM$_{10}$), discrepancies occurring only in July and September. It is worth
mentioning that the seasonal cycle of DD contribution to AOD does not follow the pattern of the total AOD
(near bell shape), meanwhile both PM$_{10}$ cycles (total and DD contribution) present more similar shapes
between them, although a main discrepancy is observed in September. The inter-annual evolution of the
DD contribution to AOD and PM$_{10}$ has evidenced a progressive decrease. This decline in the levels of
natural mineral dust aerosols can explain up to the 30% of the total aerosol load decrease observed in the
study area during the period 2003-2014. The relationship between columnar and surface DD contributions
is evident with a correlation coefficient of 0.81 for the inter-annual averages. Finally, synoptic conditions
during DD events are also analysed observing that the North African thermal low causes most of the events
(~53%). The results presented in this study highlight the relevance of the area studied since it can be
considered as representative of the clean background in Western Mediterranean Basin where DD events
have a high impact on aerosol load levels.
**Keywords:** Desert Dust Long-term Inventory; Aerosol Optical Depth; Particulate Matter; Occurrence and
Trends; Desert Dust Contribution to Aerosol Load.
**1. Introduction**
Atmospheric aerosol particles play a key role in the radiation scattering and absorption physical processes
that contribute to the Earth's radiative budget (Trenberth et al., 2009; Wild et al., 2013). Their impact  on
Earth's climate is represented by their direct radiative forcing (Haywood and Boucher, 2000; Boucher et
al., 2013), but aerosols also act as cloud condensation nuclei (CCN) modifying cloud properties and giving
rise to a set of feedback processes that constitute the indirect radiative effect (Lohmann and Feichter, 2005;
Lohmann et al., 2010; Boucher et al., 2013). All these aerosol climate effects have been enhanced due to
anthropogenic aerosol particles (mainly sulphate and carbonaceous substances) which have increased the
mean load of the world in the last century and have modified substantially the atmospheric composition
(Boucher et al., 2013). Aerosol radiative properties, such as aerosol optical depth (AOD) or single
scattering albedo (d'Almeida et al., 1991; Cachorro et al., 2000; Eck et al., 2010) are important issues to
consider when studying the impact of atmospheric aerosol on climate.
Beside the climatological aspect of atmospheric aerosols, another element to be considered is their direct
incidence on air quality (Kulmala et al., 2009; Ganor et al., 2009; Querol et al., 2013). The particulate





matter in environmental studies is mostly represented by its level of mass concentration at the surface
represented by various size fractions ($PM_{10}$, $PM_{2.5}$, $PM_1$, etc., where the subscript indicates the upper cut-
off of the aerodynamic diameter of particles and PMx is used here as a general term referring to these
fractions) and by its chemical speciation to derived specific aerosol components: sulphates, nitrates,
carbonaceous, mineral, etc.. The different properties of aerosols are linked to derived effects like the strong
adverse impact on human health (Pope 2000; Pérez et al., 2012), and ecosystems (Mahowald et al., 2010).
Desert dust or mineral aerosol is one of the main natural types of atmospheric aerosol particles, with a
strong impact on the Earth system due to its worldwide distribution and spatio-temporal variability (Goudie
and Middleton, 2006; Knippertz and Stuut, 2014; Viana et al., 2014). The injections of Desert Dust (DD)
into the atmosphere, from the two Sahara's major dust sources (Bodélé depression and Eastern Mauritania)
by different re-suspension processes, can result in high layers being transported through very large
distances (e.g., Prospero et al., 1999; 2002; Escudero et al., 2006; Engelstaedter and Washington, 2007;
2011; Knippertz and Todd, 2012; Guirado et al., 2014).
Our interest in this work focuses on atmospheric aerosol studies over the Iberian Peninsula (IP), which
constitutes a peculiar area due to the large spatio-temporal variability in aerosol properties, types and
mixing processes as a result of the contrasting influences of the Atlantic Ocean, Mediterranean Sea,
European continent and the Saharan area. Based on sun-photometer data studies, different sectors of the IP,
basically defined by its topography/geography, can exhibit different aerosol climatologies (Alados-
Arboledas et al., 2003; Vergaz et al., 2005; Estellés et al. 2007; Toledano et al., 2007a; Obregón et al.,
2012; Bennouna et al., 2013; Mateos et al., 2014a). In particular, the Sahara and Sahel desert areas are the
most important natural sources of mineral aerosols for the IP. The closeness of the IP to the African
continent intensifies the impact of desert dust events on the aerosol load, measured on the whole
atmospheric column (AOD) and at the surface (PMx). Different synoptic weather conditions and
circulation patterns define the arrival of desert dust intrusions in the IP with a different seasonal behaviour
(Escudero et al., 2005, 2006; Toledano et al., 2007b; Basart et al., 2009; Valenzuela et al., 2012; Pey et al.,
2013a; Salvador et al., 2013, 2014). These intrusions are characterized by isolated or episodic events of
short duration (around 2-3 days) but episodes in summer months are longer and more frequent in such a
way that often successive episodes are linked due to the recirculation of air masses producing feedback
processes, which give rise to a long residence time of desert dust particles in the atmosphere when there is
low precipitation (Rodríguez et al., 2002; Escudero et al., 2005). Therefore, desert dust aerosols are one of
the most important types over the IP, having an important influence on the air quality and radiative
properties and hence its detection, quantification and characterization are important research tasks.



There are different ways to approach the detection or identification of desert dust events depending on the objectives of each study. The detection depends on the different techniques used: surface measurements, remote sensing (satellite or ground-based), back-trajectories evaluation, aerosol models, or a combination of them. Air mass trajectories have been one of the first and most used techniques to identify the origin of the transport of mineral dust aerosols to different regions worldwide (e.g., Hogand and Rosmond, 1991; Prospero et al., 2002; Kallos et al., 2003; Pace et al., 2006). Although there is abundant literature about mineral dust over Southern Europe or Mediterranean areas, most of the studies about detection, characterization and/or impact of desert dust aerosols are focused on case studies or particularly strong episodes: e.g., in Italy (e.g., Meloni et al., 2007; di Sarra et al., 2011; Bègue et al., 2012), Greece (e.g., Kaskaoutis et al., 2008) or the Iberian Peninsula (e.g., Lyamani et al., 2005; Pérez et al., 2006; Cachorro et al., 2006, 2008; Guerrero-Rascado et al., 2009; Córdoba-Jabonero et al., 2011). Few studies are based on long-term datasets of desert dust using different techniques, such as sun photometers (Toledano et al., 2007b; Valenzuela et al., 2012), satellite sensors (Kaufman et al., 2005, Kaskaoutis et al., 2012; Gkikas et al., 2013; 2015), LIDAR measurements (Mona et al., 2006; 2014) or PMx data (Escudero et al., 2005; Salvador et al., 2013; 2014; Pey et al., 2013a; Rodríguez et al., 2015). As can be seen only recent studies contain long-term data sets and only in some of them the net contribution of DD is evaluated.

PMx observations provided by different networks have constituted one of the most frequent tools for the establishment of DD inventories (e.g., Escudero et al., 2005; Pey et al., 2013a; Salvador et al., 2013; 2014) in order to evaluate their contribution to PMx levels demanded by the EU directives. The EU 2008/50/EC directive (EC, 2008) on air quality establishes threshold values for the concentration of particles with aerodynamic diameter below 10 ($PM_{10}$) and 2.5 ($PM_{2.5}$) µm: an annual mean and 24 hour mean of respectively 40 µg m$^{-3}$ and 50 µg m$^{-3}$ for $PM_{10}$ and an annual $PM_{2.5}$ average of 25 µg m$^{-3}$. In this sense it is necessary to know the contribution of natural and anthropogenic components to the total aerosol load. Therefore, the contribution of mineral dust in South-Europe is important because of the linkage of $PM_{10}$ exceedances and DD outbreaks.

Once the identification or discrimination of DD African aerosols is carried out, the next task is to quantify their contribution to the total aerosol load. The evaluation of impact or contribution of DD episodes to PMx data is viable by means of a chemical speciation analysis (Rodríguez et al., 2001, 2002, 2015) but this method requires a high man power and presents poor temporal sampling. Hence, in order to avoid this expensive technique other methods have been developed taking PMx (Escudero et al., 2007; Ganor et al., 2009) and AOD data (Toledano et al., 2007b) or need to be developed. As reported by Viana et al., (2010) and taking into account more recent publications (e.g., MAGRAMA, 2013, 2015) no more than three methods are nowadays used. However, other techniques such as receptor models widely applied for surface data (Pey et al., 2013b; Belis et al., 2013) may be updated or improved for DD contribution estimates. In a





similar way columnar aerosol algorithms can facilitate the apportioning of the different aerosol types
(Dubovik et al., 2002; O'Neill et al., 2003) to the total aerosol load.
The advantage of remote sensing techniques, such as sun-photometry, for DD detection is the spectral
information recorded by their AOD measurements and given by the Ångström exponent, α. This is a
powerful tool in the identification and classification of the different aerosol types (Eck et al., 1999;
Toledano et al., 2007a) but also allows "near-real-time" processing of data by means of reasonably
sophisticated algorithms (Dubovik et al., 2002; O'Neill et al., 2003) to retrieve aerosol properties. Surface
PMx and columnar AOD quantities present notable differences between them (see Bennouna et al., 2014;
Mateos et al., 2015) and hence their DD impact can also present some discrepancies. The PMx sampling is
based on daily records (see, Aas et al., 2013) while sun photometers provide instantaneous measurements
of the columnar load but their sampling is limited to sun cloud-free conditions (Toledano et al., 2007a,b).
The usefulness of a DD inventory is that it opens the possibility of the evaluation of desert dust
contribution to the total aerosol load. However, very few studies have accomplished this task over a long
period and under a climatological perspective. To our knowledge, only the inventory of Toledano et al.
(2007b) addressed the DD contribution to AOD between 2000 and 2005 in a Spanish South-western site.
For PMx data we have found various studies, as those of more recent publication by Salvador et al. (2013;
2014) and Pey et al. (2013a). Salvador et al. (2013) reported a DD inventory and the corresponding
contribution of DD is determined over the Madrid area from 2001-2011. This study is extended to several
stations covering the whole IP by Salvador et al. (2014). Pey et al. (2013a), with the same methodology,
analysed the period 2001-2011 for $PM_{10}$ at different sites over the whole Mediterranean Basin. In these
three mentioned studies, the method followed for the evaluation of the DD contribution to PMx is different
to that for AOD. Therefore, the development of methodologies for the evaluation of DD contribution is an
open area of research.
Within this framework, the main purpose of this study is to establish a reliable inventory of DD episodes
together with the evaluation of their contribution to the total aerosol load, using both AOD and PMx data.
As an innovative aspect, this is the first time, to our knowledge, that DD events are identified by the
simultaneous use of both columnar (AOD/α) and surface (PMx) aerosol observations. The methodology is
applied over one of the atmospheric cleanest areas in Southwestern Europe, the North-central area
('Castilla y León' region) of the Iberian Peninsula, for a long time period spanning more than one decade
(2003-2014). Regarding the columnar aerosol data, the reliable measurements performed within the
Aerosol Robotic Network, AERONET (Holben et al., 1998), are considered. For the study area, the only
available aerosol long-term data set is recorded in Palencia site (41.9° N, 4.5° W, and 750 m a.s.l.).
Regarding surface aerosols, the high quality particulate matter data recorded by European Monitoring and
Evaluation Programme (EMEP) network are considered in the nearby Peñausende station (41.28°N,





5.87ºW, and 985 m a.s.l.), with similar background conditions to Palencia. In this way, the usage of these
two worldwide extended networks ensures the feasibility of implementing the proposed method in other
regions. The long-term inventory described hereafter has been employed to establish the main
characteristics of the DD episodes in North-central Spain: their climatology, inter-annual behaviour, trends
for the number of episodes and associated days, and occurrence under different synoptic scenarios. In
addition, the evaluation of the DD contribution to the total mean values of AOD and PMx is also addressed
over the period investigated under a climatological and inter-annual perspective, which emphasizes the
correlations between both quantities.
Section 2 describes the region of study and the datasets used. The methodology followed in the DD
identification and in the evaluation of its contribution is presented in section 3. In subsections 4.1-4.3, the
seasonal cycles and inter-annual evolution of DD events, dusty days, and DD contribution to AOD and
PMx are deeply investigated. Subsection 4.4 provides an estimation of the uncertainty of this method and
subsection 4.5 describes the synoptic scenarios leading the arrival of DD episodes to North-central Iberian
Peninsula. Finally, Section 5 sums up the main findings obtained in this study.
**2. Sites of measurements and database**
*2.1. AERONET network and AOD/ Ångström exponent database*
The main database for this study contains the instantaneous values of AOD obtained for 440 nm
(henceforth AOD for shake of clarity) and Ångström exponent (α for the 440-870 nm range) measured in
Palencia site of the AERONET-Europe network (see Figure 1 and Table 1). The instrument used to obtain
these data is a CIMEL CE318 radiometer which measures under clear sky conditions and each 15 minutes.
The raw measurements in all sky conditions (level 1.0 of AERONET criteria), the cloud screened data
(level 1.5), and the high quality processed data in the level 2.0 are used in this study. Lower quality levels
are of great help to reliably determine the duration of each DD episode, in particular, when the event is
mixed with cloudiness.
Aerosol measurements are available in the Palencia site since 2003, one of the longest series of aerosol
optical measurements in the Iberian Peninsula (Mateos et al., 2014a). The number of available days for
every year in this study can be seen in Table 1. The standardization protocols of AERONET demand pre-
and post- calibration of the instruments after a field measuring period of 12 months, which helps to assure
the quality of the obtained data and associates an uncertainty for $AOD_{440nm}$ of ±0.01 and for the derived α
parameter of about ±0.03 (see, e.g., Toledano et al., 2007a).





The Palencia site is placed in the autonomous region of "Castilla y León" in the North-central Iberian
Peninsula, which is also known as the "Castilian plateau" with an average altitude of ~800 m.  This region
is the third less-populated community in Spain due to its big surface (94,193 km$^2$) and its low population
(2.543.413 inhabitants registered in the census in 2012), with a population density just a bit higher than 27
inhabitants per km$^2$. Palencia is a small city (100.000 inhabitants) placed in the north of "Castilla y León"
but the measuring site is located outskirts being surrounded by rural areas, removed from of big urban and
industrial centres. Hence, this area exhibits an exceptionally clean atmosphere and aerosol observations are
representative of the background conditions for the whole region. Therefore, desert dust intrusions are
observed since they notably modify the background aerosol properties.
In order to fill the gaps in the Palencia AOD database, the site "Autilla" (42.00ºN, 4.60ºW, and 873 m
a.s.l.) close to Palencia (7 km apart) has been used (see for details Bennouna et al., 2013). This site is being
used by GOA (Grupo de Optica Atmosférica) as the calibration platform for Cimel sun photometers within
the AERONET-EUROPE infrastructure and also works as a routine measurement site. Under these
considerations, the columnar aerosol data series used in this study is consistent and allows one to perform
the inventory of DD events in this region.
***2.2. EMEP network and PM database***
Daily PM$_{10}$ and PM$_{2.5}$ measurements provided by the EMEP network constitute the second core database
used to carry out this study (see Figure 1 and Table 1). This network has the objective of regularly
providing qualified scientific data to the interested organizations in order to analyse and assess the
transboundary transport and emission of pollutants (e.g., Aas et al., 2013). This is the objective of the
LRTAP (Long-Range Transboundary Air Pollution) convention that establishes a framework for
cooperative action for reducing the impact of air pollution. Using the PM$_{10}$ and PM$_{2.5}$ data, the particulate
matter associated with coarse particles (PM$_{2.5-10}$) can be determined by subtracting these quantities.
PM$_{10}$ and PM$_{2.5}$ data belonging to the EMEP site of Peñausende have been used for the detection of days
with DD load in correspondence with AOD-α data of the Palencia site and for the evaluation of their
contribution to the total PMx levels. However, in order to better detect the DD episodes that arrive to our
study area, two nearby stations (Campisábalos, 41.28ºN, 3.14ºW, and 1360 m a.s.l., and Barcarrota,
38.48ºN, 6.92ºW, and 393 m a.s.l., see Figure 1) are taken as complementary sites. All these three sites are
placed in rural areas where background values are measured and the detection of Saharan desert dust
intrusions is also possible. Among all Spanish EMEP sites, these two complementary sites have not been
randomly selected, since their geographical locations help us to have more information about the path
followed by the intrusion before arriving in our study area. The arrival direction can be established from





the west (by the Peñausende site), South-west (by the Barcarrota site) and South-east (by the Campisábalos
site).
The available time period for the PMx data starts in 2001, but the same period (2003-2014) used in the
$AOD_{440nm}$ is considered. It is important to emphasize here that in spite of the distance between Peñausende
and Palencia sites (~100 km), the absence of any large landforms between them together with their
atmospheric and background conditions make possible the joint discrimination and evaluation of these
observations of AOD and PMx for the detection of DD intrusions. Furthermore, a deep analysis about the
air masses at Peñausende and Palencia sites (not shown here) has been carried out, corroborating that the
geographical distance between them is negligible in the analysis of regional quantities.
The use of $AOD_{440nm}$, $\alpha$, $PM_{10}$, and $PM_{2.5}$ observations provides a complete and detailed database to carry
out an analysis in terms of aerosol load and particle size, both at the surface and in the whole atmospheric
column. Table 1 presents a detailed description of the annual sampling of each database. Overall, the PMx
temporal sampling is larger than for AOD. Particularly, the year 2003 presents the lowest AOD sampling
because of the gaps at the start of the sun photometer measurements.
*2.3. Ancillary information*
With the aim of performing a more accurate evaluation and discrimination of days that constitute a desert
dust intrusion, ancillary information is taken into account. Air mass back-trajectories arriving at Palencia at
12.00 UTC have been calculated with the HYSPLIT model (Hybrid Single-Particle Lagrangian Integrated
Trajectory), version 4 (Draxler et al., 2014; Stein et al., 2015). Due to the fact that desert dust aerosols can
be transported to altitude levels higher than the boundary layer, back-trajectories have been calculated for
three heights (500, 1500 and 3000 meters above ground level) and analysed 5 days back in time (120 h),
assuming the model vertical velocity in the calculations. The meteorological database used as input for
HYSPLIT is the Global Data Assimilation System (GDAS) dataset. These three levels are very usual in
this type of studies to represent the air masses near the surface, in the boundary layer and in the free
troposphere, in order to follow the vertical transport of aerosols.
Valuable information about cloudiness is obtained from MODIS (Moderate Resolution Imaging
Radiometer) rapid response imagery products (https://earthdata.nasa.gov/data/near-real-time-data/rapid-
response). In addition, GIOVANNI (Geospatial Interactive Online Visualization ANd aNalysis
Infrastructure) MODIS AOD aerosol maps (http://giovanni.gsfc.nasa.gov/giovanni/) and those provided by
the AERONET website (http://aeronet.gsfc.nasa.gov/cgi-bin/bamgomas_interactive) are used to determine
the extension and path followed by the mineral dust air masses for a DD identified event. The NAAPS



Global Aerosol model (Navy Aerosol Analysis and Prediction System; available at
http://www.nrlmry.navy.mil/aerosol/) is also used.
As it has been analysed in previous studies (Escudero et al. 2005; Toledano et al. 2007b) desert dust
intrusions over Spain take place under certain synoptic scenarios. For evaluating these scenarios, the
geopotential height at 700 hPa and mean sea level pressure are required. Through the Earth System
Research Laboratory of NOAA (National Oceanic and Atmospheric Administration), the plots of
atmospheric quantities can be obtained for the selected days and one possible scenario among the four
possibilities proposed by Escudero et al. (2005) is identified.
**3. Methodology**
*3.1 Detection of desert dust episodes*
This study is based on instantaneous $AOD_{440nm}$ and $\alpha$ values, as well as daily $PM_{10}$ and $PM_{10}/PM_{2.5}$ ratio
data. The method for the detection of Desert Dust (DD) intrusions is a manual inspection of the evolution
of these four quantities together with the origin of the air masses at the three levels of altitude and the
auxiliary material of AOD MODIS maps, aerosol models and synoptic scenarios. The methodology for
detection is similar to that applied in Toledano et al., (2007b) with the added information of PMx data, and
not so much different from that used in other studies also based on a set of different observations (Escudero
et al., 2005; Pace et al., 2006; Kalivitis et al., 2007; MAGRAMA, 2013). The difference between these
methods lies on the weight played by each quantity, or the way to analyse the information. For example in
our case the AOD-PMx data is the first and primary information, but in other studies the key variable is the
origin of the air masses (Pace et al., 2006; Valenzuela et al., 2012). Meteorological products and forecast
aerosol models can also be used for this task (MAGRAMA, 2013). Although automatic methods can be
applied in the DD identification, a visual inspection should be performed to corroborate each classification.
This study has been carried out as a year by year service to the "Consejería de Medio Ambiente" of the
Autonomous Community of 'Castilla y León' by means of two Research Programmes from 2006-2013
about the "Discrimination, characterization and evaluation of desert dust outbreaks over 'Castilla y León'
region". These programmes have the purpose of accomplishment programmes aim to help accomplish the
"Environmental Quality Improvement Policy" of the EU by the National and Regional Governments. The
experience gained with this year by year identification gives rise to the final DD inventory presented in this
study.
Certain thresholds have been established in order to identify those conditions which are separated from the
background over the studied area. Hence, these choices are based on the aerosol climatology of the site. For





Palencia, mean $AOD_{440nm}$ is 0.13 with a standard deviation (std) of 0.09 and $\alpha=1.27 \pm 0.36$. Consequently,
DD intrusions are firstly detected using the following thresholds: $AOD_{440nm} \geq 0.2$ (AOD Criterion in Table
1), which is the mean plus the standard deviation approximately, and $\alpha \leq 1.0$ for instantaneous values. If a
day has about 50% of these data, the day is considered as a dusty day and these events are called "pure
desert aerosols", denoted by (D). However, days with instantaneous values of $AOD_{440nm} \geq 0.2$ and $1.0 \leq \alpha$
$\leq 1.5$ may also present desert dust aerosols but mixed with other types (indicated by the high $\alpha$ value),
which is corroborated by the desert origin of air mass backwards trajectories, satellite images, and model
forecasts. This type of days, named MD (Mixed Desert), may either be a part of an intense event (generally
of D type) or form by themselves a low-moderate intensity event. Previous studies have also stated that DD
intrusions in the Mediterranean Basin can present moderate AOD associated with large $\alpha$ values (e.g., Pace
et al., 2006; Tafuro et al., 2006).
The limit of $\alpha < 1$ for identifying coarse particles has been established by previous studies in different areas
(e.g., Eck et al., 1999; 2010; Dubovik et al., 2002, Meloni et al., 2007), making this threshold suitable for
our study area. It is important here to emphasize that the $\alpha$ parameter allows a more fine identification of
desert dust events, mainly those of low intensity with a less desert character (the larger the AOD of a desert
event, the lower $\alpha$), generally those mixed with other type of aerosols which are not accounted for in many
DD studies (e.g., Gkikas et al., 2013). DD events of low intensity are more difficult to detect and hence it is
more difficult to evaluate their contribution or impact on the AOD and PMx daily values.
The criterion for the $PM_{10}$ values is selected from our investigations (Bennouna et al., 2014; Mateos et al.,
2015) and previous results (see, e.g., Querol et al., 2009; 2014), giving a threshold of $PM_{10} \geq 13$ µg m$^{-3}$ for
the daily values of the particulate mass concentration (PM Criterion in Table 1). The mean value of $PM_{10}$
over the period 2003-2014 is $10\pm9$ µg m$^{-3}$, thus 13 is the mean plus one third of the standard deviation.
However, this value alone must be taken as a "warning value" and not as a true threshold, in the sense that
this value alone does not define a dusty day. This $PM_{10}$ value could seem very low but we must note here
that all these values are manually supervised by the expert-observer who will take the final decision of the
inclusion or not of a dusty day, having in mind all the available information given by these data and all the
complementary material.
Another problem to be considered in the identification of a dusty day is when AOD and $PM_{10}$
measurements show different information, e.g., AOD indicates desert dust presence but $PM_{10}$ does not. It
must be taken into account that $PM_{10}$ quantity does not necessarily follow the same temporal behaviour as
the AOD and possible delays due to deposition can occur. Desert dust events can reach the IP at high
altitude layers (e.g., above 2000 meters, Gkikas et al., 2015), and dry deposition can last various
hours/days. Assuming an average speed of deposition of around 0.6 cm s$^{-1}$ (Zender et al., 2003), DD



particles can remain up to two days after an episode ends (Escudero et al., 2007). Hence, the delays
between air masses and $PM_{10}$ are very variable (e.g., Kalivitis et al., 2007; Pey et al., 2013a). Therefore, as
mentioned AOD and $PM_{10}$ observations must be consider as complementary information to detect mineral
dust aerosols.
It is worth mentioning here that in the general detection of both intensity and duration of a particular event,
DD causes an increase in the $AOD_{440nm}$ and $PM_{10}$ values, which then surpass the corresponding threshold,
together with a decrease in the $\alpha$ and $PM_{2.5}/PM_{10}$ values given rise to an increase of the mean size of the
particle distribution. The duration of each intrusion can be established, since the background values are
recovered when the event finishes. The central or most intense days of each event are easy to detect due to
the large increment of aerosol load with large particles ($\alpha$ even close to zero) but low to moderate events
are more difficult to detect. Although the events vary in their nature, the first and last days of a DD event
show a low or moderate signature of mineral dust particles because of its mixture with other aerosol types
(clean continental aerosol in our study area), with the exception of the strong DD events which generally
have a notable impact on the aerosol load levels since the first day.
In the inspection of the instantaneous columnar dataset, non-reliable records are identified and removed
due to their high dispersion likely attributed to cloud contaminated conditions. To corroborate some critical
decisions, the ancillary information (see section 2.3) constitutes a key point of the methodology. For
instance, the verification of cloudy conditions can be supported by means of a comparison between AOD
instantaneous data from different levels, 1.0 (all-sky conditions) and 1.5 (cloud-screened data), and the
visualization of cloud systems in MODIS true colour and cloud product images. If there are signs of cloud
presence, instantaneous AOD data are carefully checked to discern between non-valid data and a DD
intrusion.
Once aerosol load measurements for a certain day indicate the likely classification as a dusty day, the air
mass back-trajectories (calculated as described in Section 2.3) are visualized in order to check if the origin
of the path or the path followed crosses the North-African region and/or its surroundings. Therefore, the air
mass back-trajectory analysis and the geopotential maps (establishing a particular synoptic scenario, see
Sections 2.3 and 4.5) lead to the final decision with respect to a DD event day classification, even in those
days showing cloudiness. Finally, to help in the understanding of the general situation about the
geographical distribution of the aerosol plumes, AOD MODIS maps and NAAPS forecasts are also
inspected. The consistency of the information used provides a reliable identification of the DD event.
It is worth mentioning here that the final decision to include a day as D or MD is made by the human-
observer with all the available information at hand. Perhaps this methodology is not the most adequate to



apply for a big area with a high number of stations and long-term databases, but it is necessary for
developing methodologies, because it will allow validation of other more automatic methods (e.g., those
only based on threshold criteria), most of them using satellite observations (e.g., Gkikas et al., 2013, 2015).
### *3.2 Evaluation of desert dust contribution to total AOD and total PM$_{10}$ concentration*
Once the DD inventory is established, the evaluation of DD contribution can be addressed at seasonal and
annual scales. Following Toledano et al., (2007b), the contribution of the DD events to AOD can be
obtained as the difference of the multi-annual monthly means considering all days and the corresponding
value without including the desert dust cases. This procedure was also used for PM$_{10}$ data in a 3-year
evaluation of net DD contribution in several sites at the Mediterranean Basin (Querol et al., 2009). In this
study, the annual cycle of the DD contribution to AOD/PMx is evaluated with this same methodology over
the entire period 2003-2014, using the DD event days classified in the inventory. Furthermore, the relative
DD contribution to AOD/PMx can be obtained normalizing with respect to the total AOD/PMx value.
Regarding the seasonal evaluations, the classification is used as follows: winter (DJF), spring (MAM),
summer (JJA), and autumn (SON). Analogously, the yearly AOD/PMx means excluding dusty days are
subtracted from the yearly AOD/PMx means for all days for obtaining the DD contribution at annual time
scale.
This method assumes the entire daily aerosol load (both surface and columnar) due to DD aerosols, being
included the contribution of regional background aerosols. Thus, suitable time scales for this kind of DD
contribution calculation are the annual and climatological monthly means. Those evaluations for every
single day or month can be addressed using other methods, for instance the determination of percentile 40
to evaluate the background conditions that are subtracted from PMx levels (Escudero et al., 2007). This
method has been taken as the standard by the European Commission for the evaluation of DD contribution
(Viana et al., 2014; MAGRAMA, 2013, 2015).
The methodology used in this study leads to lower uncertainty in the annual cycle evaluation since there is
good data coverage for the multi-annual monthly sampling, but higher uncertainty for a yearly evaluation.
The inclusion or not of an uncertain DD event (e.g., very contaminated with clouds where cloud optical
depth is assigned to aerosol AOD) can substantially modify the corresponding yearly mean as it has been
shown in Bennouna et al. (2014). This source of uncertainty must be considered in the temporal trend
evaluation. It is not easy to establish an adequate methodology to evaluate the DD aerosol contribution
(Viana et al., 2010) and much less with its corresponding associated error. A further investigation is
necessary about this subject. A discussion about the uncertainty of our approaches in the DD identification
and in the evaluation of DD contribution to aerosol load can be found at the end of the results section.




## 4. Results and discussion

### 4.1 Evaluation of the number of episodes and days: annual cycle and year-to-year variability

### 4.1.1. Mean Evaluation of the number of episodes and dusty days

The inventory of desert dust intrusions includes: information on each episode and its associated days; the daily mean AOD, α, $PM_{10}$, and $PM_{2.5}$; cloudiness, synoptic scenarios, and air mass origin at the three altitude levels mentioned above. Tables 1 and 2 show the information used to classify DD events and the main statistics for this inventory, respectively.

The $PM_{10}$ sampling presents the best coverage of the measuring time period with a 93.1% of the days, AOD is available 67.2% of the time, and the coincident sampling is available 63.2% of the time. As can be deduced from Table 1, the majority (51.3%) of the DD event days composing the inventory are noticeable in both AOD and $PM_{10}$ datasets. However, 46.3% of the total detected days are over the required thresholds only in one quantity (AOD or $PM_{10}$). This is the great advantage of the proposed inventory. The reasons behind this 46.3% (19.1% only with AOD and 27.2% only with $PM_{10}$) are due to delays between columnar and surface levels related to deposition phenomena and the lack of AOD or $PM_{10}$ measurements. Finally, a smaller number of cases (2.4%) are identified as dusty days using the ancillary information when AOD and $PM_{10}$ data are not available.

The smaller coverage of AOD is not a major handicap in DD detection. There are several years (2004, 2007, 2008, 2010, 2011, and 2012) with more DD days detected only by AOD than only by $PM_{10}$, in spite of the smaller AOD sampling (between 53 and 103 days less per year). However, years 2003 and 2006, which present less than 200 daily AOD data, require the use of $PM_{10}$ in order to better identify DD intrusions.

As reported in Table 2, during 2003-2014, a total number of 152 episodes have been identified, composed of 419 days. Among them, 243 have been classified as days with desert aerosols (D) and 176 with mixed aerosols (MD). Overall, this means 13 episodes and 35 days per year with desert dust intrusion, representing the 9.6% of the days each year. The duration of DD episodes is very variable, ranging from 1 to 13 days, but a value of 2.7 days is obtained as the mean episode duration. Due to the high variability of these intrusions it is difficult to distinguish when an event has ended or its intensity has simply fallen below our threshold. During summer, the recirculation of air masses in the IP is very frequent and the DD episodes are subject to large variations. We have considered separate DD episodes when there is, at least, one day that does not meet the DD requirements between two DD episodes.




Our percentage of dusty days of 9.6% is lower than that reported by Salvador et al. (2013), which is around
18% (18 episodes and 65 days per year), who analysed DD intrusions over the central Iberian Peninsula
(Madrid area) between 2001 and 2008. This large difference between two nearby areas (separated by ~200
km) can be explained by the different time periods considered and the existence in between of a high
mountain range (*Sistema Central*), up to 2400 m (a.s.l.). For the North-eastern area of the Iberian
Peninsula, Escudero et al. (2005) reported 15% of DD intrusions (16 episodes and 54 days per year) in the
period 1996-2002, and Pey et al. (2013a) obtained 17-18% between 2001 and 2011. In the comparison with
our results, we must note that not all DD episodes have a net impact on PMx levels because of the
mentioned characteristics of transport and deposition of DD intrusions. The contribution of these episodes
to PMx and AOD will be analysed in the next subsections.

### 4.1.2. Annual cycle of the number of episodes and days

The annual cycles of the number of episodes and number of days with DD conditions are presented in
Figure 2. In general, the seasonal pattern along the year followed by the number of episodes (Figure 2a)
and dusty days (Figure 2b) is similar with a significant increase of the DD occurrence in March (13 events
and 34 dusty days), a weak fall of DD event days in April (14 events and 25 dusty days), a notable
increment between May and September (around 17 events and 53 dusty days per month), and a progressive
decline to the minima in November and December. The number of episodes and event days peaks in June
(20 events and 65 dusty days). A noteworthy feature of this figure is the non-expected local minimum in
the number of DD episodes in July (15 episodes in 2003-2014) which is shifted to August in the number of
DD event days (46 dusty days in 2003-2014). Figure 2b is similar in shape to that reported by Salvador et
al., (2013) with the exception of September and also similar to that of Escudero et al. (2005) with the
exception of October. Concerning the two types of DD conditions distinguished in our inventory, D type
controls the annual cycle in the March maximum and April minimum, while MD controls the evolution
between August and October.
Some features mentioned above regarding the seasonal behaviour of DD events for the North-central
Spanish region are also observed for other areas of the IP. For instance, the March maximum and April
minimum are common features in South-western (Toledano et al., 2007b; Obregón et al., 2012), North-
eastern (Escudero et al., 2005), and Central (Salvador et al., 2013) Spain. In spite of the different time
periods and methodologies employed in the DD event identification, its impact over (almost) all the Iberian
Peninsula seems to follow the same pattern with the two maxima, one at late-winter/early-spring (March)
and the other in summer, and the accentuated minimum of winter. Several minor discrepancies are found
for the rest of the year; for example, the maximum number of DD event days during summer months with a
local minimum (in July) between them seems to be a characteristic of the annual cycles of the South-





western (Toledano et al., 2007b) and North-central areas. Conversely, the eastern region does not show this
behaviour and presents a local maximum in October (Escudero et al., 2005; Pey et al., 2013a). These
results confirm that different areas have different aerosol properties in the IP (Mateos et al., 2015).
*4.1.3. Interannual Variability and trends of the number of episodes and days with DD*
The year-to-year variability for both number of episodes and days is reported in Table 2 and illustrated in
Figure 3. A large inter-annual variability is observed, more accentuated for the evaluation of the number of
DD event days (Figure 3b) with an apparent decreasing trend during the analysed period. A large number
of dusty days is reported during the first five years (2003-2007), being the maximum in 2006 (68 days).
Beyond 2007 there is a decline of DD event days up to 19 days in 2010, and a small upturn is observed in
2011, 2012, and 2014 with a sharp reduction in 2013. The lowest occurrence of DD events occurred in
2013 with only 7 episodes lasting 15 days (4.11%). On the other side, the largest occurrence is registered in
2006 with 17 episodes composed of 68 days of dust intrusion (18.6%). Nevertheless, the number of
episodes and the number of days are not directly linked. For instance, 12 episodes are observed in both
2007 and 2010 but the former registered 44 days of intrusion whereas the latter just 19 days. Furthermore,
even though 2006 is the year with the highest occurrence of days, it is not linked with the most intense
events. Both $AOD_{440nm}$ and $PM_{10}$ means (for DD days) are lower in 2006 than for previous years, in which
a smaller occurrence of DD conditions is observed. The minimum load during DD days is registered in
2013 for the $AOD_{440nm}$ (0.18) and in 2009 for the $PM_{10}$ (16 µg m$^{-3}$), while the maximum occurred in 2004
(0.33 for AOD and 30 µg m$^{-3}$ for $PM_{10}$). Concerning the two classifications of DD event days, years 2003,
2006, 2009, 2012, and 2014 are governed by D type intrusions.
To quantify the decreasing trend rates in the number of episodes and associated days, the Theil-Sen
estimator and Mann-Kendall test for significance have been used. The trends for the number of DD
episodes and days are reported in Table 3 for the yearly values. A statistically significant trend at the 95%
significance level presents a *p-value* below 0.05 (e.g., Sanchez-Lorenzo et al., 2013). The total number of
dusty days has decreased by -2.7 days per year (*p-value* of 0.02) between 2003 and 2014. This strong
change, however, does not cause a significant trend in the number of episodes which presents a rate of -
0.67 episodes per year with a *p-value* around 0.03 (~97% of significance level). These figures corroborate a
notable decrease in the DD events seen in the North-central area of the Iberian Peninsula over the past
decade. This result is in line with the findings obtained by Gkikas et al. (2013) for the whole Mediterranean
Basin using MODIS data between 2000 and 2007 and considering only very intense DD events.
*4.2. Desert dust contribution to total AOD: seasonal cycle, inter-annual variability and trends*
*4.2.1. Annual seasonal cycle*



Figure 4 and Table 4 show the annual cycle of the DD contribution (small red bars in the figure) together
with the multi-annual monthly means considering all days and only days without DD aerosols (the
difference between these two values gives the DD contribution). Overall, the mean DD contribution to
AOD is 0.015 or 11.5% in 2003-2014.
The total AOD annual cycle representing the climatology follows the well-known pattern previously
reported and explained for the Palencia site (see, e.g., Bennouna et al., 2013; Mateos et al., 2014a). To
summarize: the increasing values from January to June (where the maximum is found) with a slight
reduction in May and a decreasing trend to the end of the year, provide almost a well-defined bell shape.
Concerning the climatology with the DD episodes excluded, it preserves the pattern found before for the
general case, except for some minor discrepancies. For instance, the change between May and June is not
noticeable for the curve with the DD excluded, in contrast with the larger increment observed for the
general case.
However, the seasonal pattern followed by the DD contribution to AOD is considerably different to these
two latter curves. Two maxima are observed during the annual cycle: the first one in March (late
winter/early spring) with 0.018 or 13.4% and the strongest one occurring in summer period (June and
August), ~0.027 or ~17%. Together with these maxima, there are two local minima: in April-May (around
0.014 or 9.5%) and in July (0.018 or 12%). After August, a progressive decline of the DD contribution is
observed with the minimum in winter (December and January show similar values about 0.004 or 5.4%). It
is worth mentioning here the different characters of the two local minima occurring in April-May and July,
the former is more general of the IP (linked to the precipitation cycle) while the latter is more typical of the
Central and South-western areas. For instance, the July minimum seems to be related with the arrival of
drier air masses in the low troposphere as it is observed in the precipitable water vapour cycle (Ortiz de
Galisteo et al., 2013).
The annual cycle of DD contribution for Palencia site (representing North-central Spain) presents a similar
shape to that obtained in "El Arenosillo" site (South-western area) by Toledano et al., (2007b) for an
inventory of 6 years, from 2000 to 2005. This is an important result in two aspects, one related with the
shape of the annual cycle or seasonal behaviour and the other one related with the different contribution of
North and South areas of the IP. In relation to the geographical gradient, a quantitative difference is
observed between these two areas. The total AOD signal is clearly impacted by DD events in the Southern
Iberian coast (with relative contributions being over the 30%), while in the North-central region the DD
influence is weaker, thus a South-North decreasing gradient over the IP is observed regarding the DD
contribution to AOD values. This behaviour is well known in the IP by earlier aerosol studies based on





PMx data (Querol et al., 2009; Pey et al., 2013a; Salvador et al., 2013, 2014) but this is the first time this is
confirmed by an inventory of AOD data.

### 4.2.2. Inter-annual variability and trends

With respect to the inter-annual change of the DD contribution to AOD, Figure 5 and Table 5 show its
annual values between 2003 and 2014 (using the methodology explained in Section 3.2). In a quick-look
analysis, both total AOD and DD contributions have a significant year-to-year variability with a decreasing
trend during the period studied but with different patterns (also observed in the relative DD contribution to
AOD). The maximum DD contribution with a value of 0.033 or 21.2% took place in 2004, with also a
maximum in the total AOD around 0.15 (the mean value of 2003 is clearly impacted by the low sampling:
42.7%, compared to the 72.4% in 2004). The year 2013 presents the absolute minimum of the DD
contribution to AOD with 0.004 or ~4%, with a low contribution in 2009 too (0.006 or ~5%). There is a
weak evolution of DD contribution until 2008, although 2005 presents a marked local minimum (DD
contribution to AOD around 0.016 or 11%). There are years with simultaneous decreases (2008, 2009,
2013) or increases (2011, 2014) of both total AOD and its DD contribution, but in other years they present
the opposite behaviours (2005 and 2006). The line illustrating the evolution of the relative DD contribution
to AOD highlights the minima of 2013, 2009, and 2005 and the maxima of 2004 and 2012. The high inter-
annual variability can be explained by the typical variability of the different African source-areas and
associated emission processes together with the atmospheric conditions and transport patterns of  DD
aerosols that can reach the Iberian Peninsula (Prospero et al. 2002; Kaufman et al., 2005; Escudero et al.,
2006; Knippertz and Todd, 2012; Salvador et al., 2014).
The temporal trends in total AOD and in the DD contribution to AOD are also evaluated and shown in
Table 3. The decrease of the total AOD in the Palencia site in 2003-2014 is -0.006 AOD-units per year
(with a *p-value* <0.01) or -4.6% per year, which is in line with previous findings for the same site by
Bennouna et al., (2014) and Mateos et al., (2014b) for shorter periods. With respect to the DD contribution
to AOD, a rate of -0.0019 AOD-unit per year (*p-value* = 0.02) or -11.2% per year is calculated. Therefore,
this rate represents the 30% of the total AOD decreasing trend. Hence, the natural decrease of DD aerosols
has notably affected AOD levels over North-central Iberian Peninsula during the study period.

### 4.3. Desert dust contribution to PMx levels: annual cycle, inter-annual variability and trends

### 4.3.1. Annual seasonal cycle

In the same way as for AOD, the contribution of desert dust events to mean values of $PM_{10}$, $PM_{2.5}$, and
$PM_{2.5-10}$ have also been calculated. The annual cycle and the inter-annual evolution of these three quantities



and the corresponding DD contributions are reported in Tables 4 and 5 and also illustrated by Figures 6 and
7, respectively.
The DD contribution to the total $PM_{10}$, $PM_{2.5}$, and $PM_{2.5-10}$ is not usually evaluated at the same time. To our
knowledge, this is the first time that fine and coarse mode contributions are evaluated in a long-term desert
dust inventory of this type. Furthermore, the temporal trends for the inter-annual DD contributions are also
discussed. It is worth mentioning here that as $PM_{10}$ and $PM_{2.5}$ are obtained from different filters (see
Section 2.2) while $PM_{2.5-10}$ is only available with simultaneous PMx data, the data number used in the
evaluation of DD contribution for each quantity slightly differs.
According to Table 4, the mean DD contributions to PMx during the study period are 1.3 µg m$^{-3}$ (12%) for
$PM_{10}$, 0.6 µg m$^{-3}$ (9%) for $PM_{2.5}$, and 0.8 µg m$^{-3}$ (16%) for $PM_{2.5-10}$, respectively. Our findings during
2003-2014 are in line with those given by Querol et al., (2009): 2 µg m$^{-3}$ for a 3-year period (2004-2006) at
the Peñausende site. A decreasing south to north gradient of African dust contribution to $PM_{10}$ (e.g., Querol
et al., 2009; Pey et al., 2013a) is found for the North-central area of the IP. In particular, $PM_{10}$ is similar to
the averages in the North-eastern area (< 2 µg m$^{-3}$) and smaller than the values obtained in southern sites
(up to 5-6 µg m$^{-3}$). Our relative contribution is in line with the lowest values of the ranges reported by
Salvador et al. (2013) using a chemical speciation analysis in three different sites in sites near Madrid.
The total $PM_{10}$ annual cycle (see Figure 6) is well known in the North-central area of the Iberian Peninsula
(see, e.g., Bennouna et al., 2014, Mateos et al., 2015): there are two maxima, a major one in summer and a
secondary one in early spring (considering our seasonal classification with March as part of the spring), a
winter minimum and another minimum in April. This general behaviour for the entire dataset is also
followed if the DD events are excluded. Furthermore, the evolution of DD contribution to $PM_{10}$ is very
similar to these two latter curves. The largest DD contribution is observed in March (2.2 µg m$^{-3}$ or 20%)
and summer months, June to August (~2.3 µg m$^{-3}$ or ~17%). The months of April and May (~0.9 µg m$^{-3}$ or
~9%) display a notable decrease with respect to March. After summer, there is a sharp fall in September
(1.2 µg m$^{-3}$ or 10%) producing a local minimum, and beyond October a progressive decline leading to the
weakest effect (<8%) of the African intrusions during winter months (DJF). The maximum relative DD
contribution to $PM_{10}$ can reach 20%, which is within the range (10%-50%) observed by Pey et al. (2013a)
for the eastern Spanish coast. Comparing the seasonal cycles of DD contribution to $PM_{10}$ in the latter area
with respect to North-central Iberian Peninsula, some common features appear (March maximum,
April/May decrease, summer increase, and September drop) but a particular difference occurs in October
since in the Mediterranean coast there is a notable rise of DD contribution at the surface.



Even though both AOD and $PM_{10}$ express the aerosol load, these quantities present noticeable differences.
To facilitate the comparison of the results shown above, Figure S1 (supplementary information) shows
together the annual cycles of AOD and $PM_{10}$ total means and their DD contributions. The annual cycle of
the two quantities, total AOD and $PM_{10}$, for the complete dataset follows a similar behaviour between
August and March, with the differences in April (local PM-minimum) and May (local AOD-minimum)
being remarkable, and a different evolution in June-July. These discrepancies between these quantities lead
to a moderate-high correlation coefficient of 0.82 between AOD and $PM_{10}$, but their physical meaning is
uncertain taking into account the mentioned discrepancies in the two annual cycles. With respect to the
correlation between the seasonal cycles of DD contributions, the absolute and relative ones for AOD and
$PM_{10}$ show the most significant discrepancies in July (with a local minimum of AOD) and September
(sharp fall of $PM_{10}$). Furthermore, the maximum of March is more intense for the DD contribution to $PM_{10}$
than to AOD. The correlation factors between these quantities are moderate-high: 0.84 and 0.74 for the
absolute and relative curves, respectively.
The fine mode, represented by the $PM_{2.5}$ data, follows the same pattern as $PM_{10}$ in the total and DD
contribution curves (Table 4 and Figure 6b). The DD contribution to $PM_{2.5}$ is below 10% for most of the
year, with a mean value of ~9%.
The total coarse mode ($PM_{2.5-10}$) curve is also similar to that obtained for the total $PM_{10}$, although the mean
contribution of the DD events is 16% of the total. The DD contribution to $PM_{2.5-10}$ (Table 4 and Figure 6c)
exhibits a strong maximum in March (1.7 µg m$^{-3}$ or 33%), a reduction in April and May (around 14%),
large values in June (1.4 µg m$^{-3}$ or 25%) followed by a weak decrease in July and August (1.3 µg m$^{-3}$ or
21%), and low values in autumn and winter.
*4.3.2. Inter-annual variability and trends*
The inter-annual variations of total $PM_{10}$, $PM_{2.5}$ and $PM_{2.5-10}$ and the corresponding DD contributions to
these PMx concentrations are plotted in Figure 7 and reported in Table 5. In the shape of the DD
contribution we can distinguish two periods associated with the strong minimum of 2009. The first period
has a decreasing trend from 2003-2009 where the first four years have similar DD contributions among
them. The second period starts with a strong ascent of DD contribution from 2009 to 2012, followed by a
significant fall in 2013 and a final rise in 2014. The absolute maximum DD contribution occurs in 2006
(2.4 µg m$^{-3}$ or 21%) and the absolute minimum is observed in 2013 with 0.4 µg m$^{-3}$ or 5%, although very
similar to the value in 2009. The solid line in Figure 7 illustrating the evolution of the relative contribution
highlights the minima of 2005, 2009 and 2013 and the maxima of 2004, 2006, 2012, and 2014.





The inter-annual evolutions of the total $PM_{10}$ and AOD are very similar (see Figure S2, supplementary
information) with an excellent agreement between them represented by a correlation coefficient around 0.9
in 2003-2014. With respect to the yearly values of DD contributions to AOD and $PM_{10}$, they show a
correlation coefficient of 0.81. The agreement is also quite good for the relative DD contributions to AOD
and $PM_{10}$ (correlation coefficient around 0.7). This high agreement, extremely good during 2009-2013, is
not seen for some years. For instance, the reason behind the low DD contribution to AOD in 2006 can be
explained by the poor sampling during that year (see Table 1). So far, no reasonable explanation has been
found for the strong fall between 2004 and 2005 in the DD contribution to AOD despite the fact that total
AOD and $PM_{10}$ display the same behaviour. The DD contribution to $PM_{10}$ is notably larger than that
obtained for AOD in 2014. The high inter-annual variability of these quantities highlights the necessity of
longer time periods to assess this kind of relationships, but bearing in mind that the net contribution of DD
aerosols is represented by very low values with a high uncertainty, hence this variability is into the
expected range of change. These results are of extraordinary interest for long-term studies of columnar and
surface aerosol loads in relation to their evolution and trends for climate studies because tropospheric
aerosols have a strong regional signature and the area studied presents exceptional background conditions
representative of Western Mediterranean Basin.
The weak impact of the DD events on the $PM_{2.5}$ levels (fine mode, see Figure 7b and Table 5) is reflected
in the low relative contribution with only three years (2003, 2005, and 2006) presenting values higher than
12%. The last years of the period analysed (2009-2014) present a notable low DD contribution to $PM_{2.5}$
below 7%. On the contrary, $PM_{2.5-10}$ (Figure 7c and Table 5) presents a sharper behaviour than previous
PMx results although still following the $PM_{10}$ pattern. The starting years are the ones with the largest
contributions (around 27% until 2006) while 2013 shows the minimum values (around 5%) together with
2009 (~7%).
There is a decreasing trend of all the quantities shown in Figure 7. The general decrease of PMx levels has
been previously reported for the Peñausende site and shorter periods (e.g., Barmpadimos et al., 2012;
Bennouna et al., 2014; Mateos et al., 2014b; Querol et al., 2014) and it has been corroborated with the
temporal trends obtained in this study (see Table 3). Cusack et al., (2012) pointed out a percentage
reduction ranging between 7% to 41% in the yearly $PM_{2.5}$ from 2002 and 2010 in 11 Spanish sites. In order
to quantify the observed decrease in the DD impact, Table 3 also presents the temporal trends of the DD
contribution of $PM_{10}$, $PM_{2.5}$, and $PM_{2.5-10}$. The general decrease of $PM_{10}$ (-0.46 µg m$^{-3}$ per year, with a $p$-
$value$ <0.01) in Peñausende site for the period 2003-2014 is in line with previous studies (e.g., Querol et
al., 2014; Mateos et al., 2015). Regarding the DD contribution, the fall in the three quantities is quantified
as around -10% per year. In particular, the DD contribution to $PM_{10}$ has decreased by an absolute amount





of 0.14 µg m$^{-3}$ per year (*p-value* of 0.06) and 0.08 µg m$^{-3}$ per year (*p-value* < 0.01) for PM$_{2.5}$. The reduction
observed in the DD event days (see subsection 4.1.3) has also led to a significant fall of the total particulate
matter. Comparing the temporal trends of PM$_{10}$ DD contribution and the rate for the total quantity, the DD
impact has caused 30% of the total PM$_{10}$ decrease in North-central Spain. As expected, this percentage is
smaller (about 21%) for the PM$_{2.5}$ case. In the North-eastern region, Querol et al., (2014) showed that
crustal matter accounted for 14% of the total PM$_{2.5}$ decrease between 2001 and 2012.

### *4.4 Estimation of associated uncertainty of the methodology*

No quantification has been done about the associated uncertainties in the number of events and associated
days in most of the reported bibliography. The same happens for the uncertainty linked to the DD
contribution, which can be evaluated as a consequence of the earlier error of DD detection, but also can be
evaluated based on other assumptions. A big step took place when the proposed methodology by Escudero
et al., (2007) was taken as the official standard method. However, the 30 days moving percentile used to
establish the regional background has been changed from 30% (reported by Escudero et al., 2007) to 40%
(Pey et al., 2013a; Salvador et al., 2013, 2014). It seems apparent that this percentile may be site dependent
thus demonstrating the difficulty of this evaluation. Otherwise, it must borne in mind that a big difference
exists between the Escudero et al., (2007) methodology and that applied by us. This subsection describes a
first estimated uncertainty of using the methodology proposed in this study.
Fingerprints of each DD event day are visible on at least one of the quantities related to aerosol load
(columnar or surface) analysed in the inventory evaluation (see Section 3), plus the additional
informational of air mass back-trajectories, satellite images, and synoptic scenarios. Usually, several of
these variables simultaneously corroborate the DD presence, especially due to the low background values
that characterize our region. Therefore, the thorough inspection of all the information provided by different
sources at the same time causes the error in the DD identification to be minimal. From our experience
during these 12 years of data, we consider that possible error sources can be, mainly, the following: gaps in
the data series, classification or not of a day when the aerosol load is close to the threshold values, and
uncertainty of the instrumental techniques and the ancillary tools. Therefore, we can estimate that about 3-5
days per year could be missed in the annual sum of dusty days, so the associated relative uncertainty,
considering the average of 35 DD event days per year, is ~9-15%. This estimation gives a realistic range
for the error associated with this methodology of visual inspection. The 5 days per year uncertainty (or
15%) can overestimate the real error, but even this percentage can be considered as acceptable as the
maximum average error. Regarding the sum of dusty days in the seasonal cycle, the same range of error
can be assumed in every monthly inter-annual value.



The possible of missing these few days with DD fingerprints (~3-5 per year and per inter-annual month)
leads to an uncertainty in the evaluation of DD contribution to AOD values. Hence, to quantify the
uncertainty in the seasonal cycle of the DD contribution to AOD each inter-annual monthly database is
extended adding 9% of DD event days. For these "extra" days the AOD is assumed as the mean value
during the DD events in that month. For instance, four days are added in June with a mean AOD of 0.27
and one day is added in January with $AOD_{440nm} = 0.18$. The DD contribution is calculated for this case,
evaluating the differences with those values shown in Section 4.2 (from the original database). The results
show a small change in the DD contribution to AOD, always below 0.002. For instance, for June the
relative uncertainty caused by the added days is 6.7% (the absolute DD contribution for the original
evaluation is 0.027). However, those months with less absolute DD contribution to AOD cause a relative
difference between 15% and 20% (such as January and December). Overall, the mean uncertainty is 0.0013
or 9.7%. The same procedure is applied for the inter-annual DD contribution to AOD. On average, the
inclusion of 9% DD extra days causes an uncertainty of 0.0014 or 8.3%. If the assumption of missing 3
days per year is even enlarged to 5 days per year, the uncertainties caused on the DD contribution to AOD
values only increase up to 14%. Hence, the reliability of the method followed here is demonstrated.
In the same way, the study of the uncertainties of the DD contribution to $PM_{10}$ is also addressed with the
same method (adding 9-15% extra DD event days). The results for $PM_{10}$ indicate a mean uncertainty of
0.1-0.13 $\mu g\ m^{-3}$ or 8-14% in the evaluation of both annual cycle and inter-annual evolution. This relative
uncertainty can be extrapolated to the $PM_{2.5}$ and $PM_{2.5-10}$ DD contributions.

### *4.5 Analysis of the synoptic scenarios during desert dust episodes*

Using the ancillary information used in the final choice of the DD identification, the synoptic scenarios that
favour the arrival of air masses originated in the north of Africa are also studied. These scenarios are those
defined and described by Escudero et al. (2005): via the Atlantic arch (NAH-S), directly from North-Africa
by a deep low pressure (AD) or by a convective system (NAH-A), and from the Mediterranean area
(NAD). Overall, the geographical positions and heights of the high and low pressure systems produce the
mineral aerosols to reach the IP. Figure 8 presents the annual cycle and inter-annual variability of the
number of episodes associated with each synoptic scenario. The synoptic scenario of each episode has been
established considering all the daily meteorological maps during the episode.
The synoptic scenario analysis of the DD events (see Figure 8a) has shown a predominance of the NAH-A
(81 out of 152 episodes), in particular, during the warm season (from May to October). This scenario
corresponds to a *North African High Located at Upper Levels*, produced by intense solar heating of the
Saharan desert. These air masses present large DD loads which can arrive at high altitudes (up to 5 km





a.s.l.). In our study region, the NAH-S (*North Africa High Located at Surface Level*) scenario governs (38
out of 152 episodes) the DD intrusions between December and April (being also significant in October)
and produces transport in the lower atmospheric levels (generally below 1 km a.s.l.). The AD scenario
(*Atlantic Depression*) plays a minor role (24 out of 152 episodes) but with an influence confined between
February and May, September, and November. The NAD (*North African Depression*) scenario only
presents an important contribution in March and December (9 out of 152 episodes).
The fingerprints of the evolution of these synoptic scenarios are reflected in the climatology of the DD
episodes shown in Figure 2. The rapid increase in DD events in March (see Figure 2) is caused by a larger
influence of NAH-S (3 to 5 DD events  with respect to February), the marked appearance of NAD (3
events), and a slight increase of AD (2 to 3 DD events with respect to February). The synoptic situation in
April changes and the NAD scenario almost disappears while NAH-S and AD increase their influence. The
local summer minimum in July is caused by the lower occurrence of the NAH-A conditions. Previous
studies have found this minimum for other columnar quantities, such as the vertical precipitable water
vapour (Ortiz de Galisteo et al., 2013). The absolute DD event minimum of November is caused by the
total disappearance of the NAH-A scenario.
Comparing these results with previous inventories performed in other geographical areas of the IP, the
synoptic scenario climatology presents some discrepancies. Toledano et al. (2007b) have also found for "El
Arenosillo" site (South-western IP) in the period 2000-2005 a predominance of the NAH-A conditions
during summer. However, the role played by the NAH-S seems to be minor during winter compared to the
North-central area. The DD inventory in the North-Mediterranean Spanish coast has been analysed by
Escudero et al. (2005) between 1996 and 2002. They also obtained the major predominance of the NAH-A
during summer, although the NAD scenario shows a notable impact on the DD events in May and
November. These outbreaks arriving from the Mediterranean area are also reported in the months of
February, March, and November in the "El Arenosillo" inventory.
Inter-annual distribution of DD events and the four synoptic scenarios (see Figure 8b) corroborates the
predominance of the synoptic scenario NAH-A every year. Overall, there is a mean of 7 episodes per year
due to this scenario in the North-central area of the IP, being the maximum influence in 2012 where 9 out
12 events occurred under this situation. A special feature is the simultaneous appearance of the four
scenarios only in years 2004, 2006, and 2014. The last two years of the analysed period (2013-2014) have
shown a decrease of the number of episodes that can be attributed to the absence of synoptic conditions
favouring mineral dust transport during summer (NAH-A scenario). The occurrence of the NAH-S and AD
scenarios presents high inter-annual variability but the number of DD episodes they caused is always
smaller than those caused by NAH-A. Finally, NAD conditions in our region are only relevant in 2004,





2011, and 2014 with 2, 3, and 2 events, respectively. However, this scenario plays a key role in the North-
eastern area of the IP (e.g., Escudero et al., 2005), which shows that DD intrusions arriving through the
Mediterranean area rarely reach the North-central region of Spain.
**5. Conclusions**
In this study, a methodology to obtain a reliable identification of DD intrusions is proposed and applied to
the North-central area of the Iberian Peninsula. Long-term datasets of AOD and PMx for background sites
of Palencia and Peñausende (representative of the study area) have been used as core information for the
detection of desert dust intrusions in this area during an 12-year period (from January 2003 to December
2014). The analysis of ancillary information, such as air mass back-trajectories at three altitude levels (500,
1500 and 3000 m a.s.l.), MODIS-AOD and true colour images, and meteorological maps, has been used to
precisely establish the duration of each desert dust episode, creating a reliable inventory with desert dust
episodes. Main conclusions can be summarized as follows:
1.  The simultaneous consideration of surface and columnar aerosols has been shown to be a reliable
tool in the DD identification. More than a half of the inventory has been detected by $AOD_{440nm}$ and
$PM_{10}$ data at the same time. However, each quantity is able to extend the DD detection by itself in a
large number of cases (114 and 80 out of 419 days detected by only $PM_{10}$ and AOD data,
respectively). The smaller coverage of AOD sampling is not a major handicap in this process.
2.  A total of 152 episodes composed of 419 days presented desert dust aerosols during the entire
period. The annual cycles of the number of DD episodes and days follow a similar pattern: an
increase in March, a weak fall of event days in April, a notable increment between May and
September and a progressive decline to the absolute minimum in winter, with the absolute
maximum in June and local minimum in July/August. Inter-annual variability of the number of DD
episodes and dusty days is high, ranging between 7 episodes (15 dusty days) in 2013 and 17
episodes (68 dusty days) in 2006. A temporal trend of -2.7 dusty days per year (95% significance
level) between 2003 and 2014 is obtained. Therefore, a reduction of the DD outbreaks in the North-
central area of the Iberian Peninsula is found during the period studied.
3.  Overall, the mean DD contribution to $AOD_{440nm}$ is 0.015 or 11.5%, while for the surface
concentration $PM_{10}$, $PM_{2.5}$ and $PM_{2.5-10}$ is 1.3 µg m$^{-3}$ (11.8%), 0.55 µg m$^{-3}$ (8.5) and 0.79 µg m$^{-3}$
(16.1%), respectively.
4.  The annual cycle of the DD contribution to aerosol load peaks in March, decreases in April-May,
notably increases during summer months (the AOD curve has a local minimum in July), and
experiences a progressive decline after summer (with a significant fall in September for the $PM_{10}$



curve) towards minimum values in winter. The maximum DD contribution to AOD occurs in June
and August close to 0.03, while the $PM_{10}$ maximum DD contribution reaches ~2.4 µg m$^{-3}$ in
August.
5. The inter-annual variability of the DD contribution to aerosol load is maximum in 2004 for AOD
with 0.03 and 2006 for $PM_{10}$ with 2.4 µg m$^{-3}$, and minimum in 2013 (0.004 for $AOD_{440nm}$ and 0.4
µg m$^{-3}$ for $PM_{10}$). The correlation coefficient between the DD contribution to $AOD_{440nm}$ and $PM_{10}$
yearly means is 0.81.
6. The temporal trends of the DD contribution to AOD, $PM_{10}$, and $PM_{2.5}$ have values of -0.0019 (*p-*
*value* of 0.02), -0.14 µg m$^{-3}$ (*p-value* of 0.06) per year, and -0.08 µg m$^{-3}$ (*p-value* $< 0.01$) per year in
the analysed period, respectively. This decrease of the levels of natural mineral dust aerosols
represents around the 30% of the total aerosol load decrease shown by AOD (columnar) and $PM_{10}$
(surface) in 2003-2014. This decrease is around 20% for the $PM_{2.5}$ case.
7. DD outbreaks have mainly reached the North-central Iberian Peninsula directly from North-Africa
by a convective system (NAH-A synoptic scenario), with clear predominance in the summer
months. The NAH-S (via the Atlantic arch) and AD (directly from North-Africa by a deep low
pressure) scenarios present a variable influence thorough the year, while the NAD (from the
Mediterranean area) conditions are only important in March and December.

The proposed inventory is the first one based on long-term AOD-PM data series. The use of worldwide
networks (EMEP and AERONET) ensures that this method can be implemented in other regions with
background aerosol observations, as long as nearby PMx and AOD measurement sites in clear remote
(background) locations are analysed.
With careful inspection of all the information, the inventory can be a useful tool to develop and validate
automated methodologies. The comparison between different methodologies will allows a more reliable
estimation of uncertainties in DD detection and its contribution to total aerosol load. Future studies based
on this inventory will be focused on a global characterization of microphysical and radiative properties of
desert dust including the evaluation of its radiative forcing over the study region. Therefore, these results
are useful for assessing regional climate change studies linked to atmospheric aerosols because of the
excellent clean background conditions of the area, which may be considered as one of the few sites/areas in
Southwestern Europe with these conditions.

**Acknowledgements**





The authors are grateful to Spanish MINECO for the financial support of the FPI grant BES-2012-051868
and project CGL2012-33576. Thanks are due to EMEP (especially to MAGRAMA and AEMET) and
AERONET-PHOTONS-RIMA staff for providing observations and for the maintenance of the networks.
The research leading to these results has received funding from the European Union Seventh Framework
Programme (FP7/2007-2013) under grant agreement Nr. 262254 [ACTRIS 2]. We also thank "Consejería
de Fomento y Medio Ambiente" for their support to desert dust studies in "Castilla y León" region, as well
as "Consejería de Educación of Junta de Castilla y León" for financing the project (VA100U14).

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





Tables

Table 1. Yearly sampling of AERONET and EMEP databases (all days) used in this study. Yearly number
of days in the DD inventory (desert dust event days) identified by criteria of AOD, $PM_{10}$, AOD&$PM_{10}$ and
other ancillary information. The relative coverage (percentage) is also given in parenthesis. See Section 3
for further details about the criteria used.

| | 2003 | 2004 | 2005 | 2006 | 2007 | 2008 | 2009 | 2010 | 2011 | 2012 | 2013 | 2014 | Total |
|---|---|---|---|---|---|---|---|---|---|---|---|---|---|
| | All Days | | | | | | | | | | | | |
| Sampling AOD | 156 | 265 | 295 | 190 | 271 | 280 | 256 | 244 | 269 | 252 | 220 | 249 | 2947 |
| (%) | (42.8) | (72.4) | (80.8) | (52.1) | (74.2) | (76.5) | (70.1) | (66.8) | (73.7) | (68.9) | (63.3) | (68.2) | (67.2) |
| Sampling $PM_{10}$ | 343 | 349 | 340 | 347 | 349 | 333 | 341 | 347 | 347 | 319 | 332 | 335 | 4082 |
| (%) | (94.0) | (95.4) | (93.2) | (95.1) | (95.6) | (91.0) | (93.4) | (95.1) | (95.1) | (87.2) | (91.0) | (91.8) | (93.1) |
| Coincident Sampling | 149 | 256 | 279 | 183 | 259 | 255 | 243 | 238 | 256 | 219 | 200 | 234 | 2771 |
| (%) | (40.8) | (69.9) | (76.4) | (50.1) | (71.0) | (69.7) | (66.6) | (65.2) | (70.1) | (59.8) | (57.5) | (64.1) | (63.2) |
| | Desert Dust Event Days | | | | | | | | | | | | |
| Number of dusty days | 44 | 44 | 41 | 68 | 44 | 31 | 24 | 19 | 32 | 29 | 15 | 28 | 419 |
| Only AOD Criterion | 5 | 8 | 9 | 6 | 14 | 4 | 4 | 5 | 9 | 12 | 2 | 2 | 80 |
| (%) | (11.4) | (18.2) | (22.0) | (8.8) | (31.8) | (12.9) | (16.7) | (26.3) | (28.1) | (41.4) | (13.3) | (7.1) | (19.1) |
| Only $PM_{10}$ Criterion | 19 | 3 | 11 | 37 | 6 | 2 | 7 | 2 | 0 | 1 | 8 | 18 | 114 |
| (%) | (43.2) | (6.8) | (26.8) | (54.4) | (13.6) | (6.5) | (29.2) | (10.5) | (0) | (3.4) | (53.3) | (64.3) | (27.2) |
| AOD&$PM_{10}$ Criteria | 20 | 33 | 21 | 24 | 22 | 25 | 11 | 9 | 23 | 15 | 4 | 8 | 215 |
| (%) | (45.5) | (75.0) | (51.2) | (35.3) | (50.0) | (80.6) | (45.8) | (47.4) | (71.9) | (51.7) | (26.7) | (28.6) | (51.3) |
| Other Criteria | 0 | 0 | 0 | 1 | 2 | 0 | 2 | 3 | 0 | 1 | 1 | 0 | 10 |
| (%) | (0) | (0) | (0) | (1.5) | (4.5) | (0) | (8.3) | (15.8) | (0) | (3.4) | (6.7) | (0) | (2.4) |








Table 2. Main results of the DD inventory. Legend: N.E. (number of episodes), N.D. (number of days),
P.D. (Percentage of days), and M.D. (mean duration). Yearly mean values of AOD, α and PMx data of
desert dust events are also reported.

| | 2003 | 2004 | 2005 | 2006 | 2007 | 2008 | 2009 | 2010 | 2011 | 2012 | 2013 | 2014 | Total | Mean |
|---|---|---|---|---|---|---|---|---|---|---|---|---|---|---|
| N.E. | 15 | 13 | 16 | 17 | 12 | 14 | 10 | 12 | 15 | 12 | 7 | 9 | 152 | 12.7 |
| N.D. | 44 | 44 | 41 | 68 | 44 | 31 | 24 | 19 | 32 | 29 | 15 | 28 | 419 | 34.9 |
| P.D. (%) | 12.05 | 12.02 | 11.23 | 18.63 | 12.05 | 8.47 | 6.58 | 5.21 | 8.77 | 7.92 | 4.11 | 7.67 | 9.6 | 9.6 |
| M.D. (days) | 2.93 | 3.14 | 2.56 | 4.00 | 3.67 | 2.21 | 2.40 | 1.58 | 2.13 | 2.42 | 2.14 | 3.11 | 2.7 | 2.7 |
| | | | | | | | | | | | | | | |
| Mean $AOD_{440nm}$ | 0.32 ±0.11 | 0.33 ±0.16 | 0.28 ±0.13 | 0.24 ±0.08 | 0.29 ±0.11 | 0.27 ±0.10 | 0.20 ±0.05 | 0.31 ±0.11 | 0.27 ±0.09 | 0.27 ±0.10 | 0.18 ±0.11 | 0.19 ±0.09 | -- | 0.26 ±0.05 |
| Mean α | 0.98 ±0.33 | 0.92 ±0.40 | 0.95 ±0.44 | 0.88 ±0.32 | 1.17 ±0.40 | 1.02 ±0.44 | 0.92 ±0.27 | 0.91 ±0.50 | 0.90 ±0.36 | 0.83 ±0.48 | 1.22 ±0.29 | 0.63 ±0.41 | -- | 0.94 ±0.15 |
| Mean $PM_{10}$ ($\mu g\ m^{-3}$) | 28.7 ±13.0 | 30.0 ±32.7 | 29.8 ±28.5 | 21.2 ±8.0 | 19.3 ±12.0 | 21.5 ±8.0 | 16.0 ±6.5 | 25.0 ±25.8 | 21.8 ±11.0 | 23.2 ±20.4 | 16.4 ±4.8 | 22.3 ±8.8 | -- | 22.9 ±4.7 |
| Mean $PM_{2.5}$ ($\mu g\ m^{-3}$) | 14.9 ±6.3 | 14.4 ±8.9 | 14.7 ±10.0 | 12.0 ±3.7 | 10.0 ±4.1 | 13.8 ±4.4 | 8.5 ±5.0 | 10.6 ±9.4 | 8.7 ±3.1 | 7.9 ±3.3 | 8.5 ±3.7 | 8.1 ±2.6 | -- | 11.0 ±2.8 |
| Mean $PM_{2.5-10}$ ($\mu g\ m^{-3}$) | 13.9 ±9.1 | 15.9 ±25.1 | 14.7 ±20.9 | 9.2 ±6.5 | 9.3 ±8.4 | 7.6 ±5.8 | 7.6 ±3.1 | 14.4 ±16.5 | 12.5 ±8.1 | 15.5 ±17.7 | 7.6 ±4.7 | 14.6 ±8.0 | -- | 11.9 ±3.4 |








Table 3. Temporal trends (Theil-Sen estimator), p-value and confidence interval ([i1, i2]) given by the
considered quantities for all days and for the contribution of DD. For the DD inventory the number of
episodes and DD event days are also included.

| | Quantity | Trend | p-value | i1 | i2 | Trend units | Trend (% per year) |
|---|---|---|---|---|---|---|---|
| **ALL DAYS** | AOD | -0.006 | <0.01 | -0.009 | -0.003 | AOD-units per year | -4.6 |
| | $PM_{10}$ | -0.46 | <0.01 | -0.66 | -0.30 | $\mu g\ m^{-3}$ per year | -4.5 |
| | $PM_{2.5}$ | -0.38 | <0.01 | -0.49 | -0.30 | $\mu g\ m^{-3}$ per year | -6.3 |
| | $PM_{2.5-10}$ | -0.07 | 0.19 | -0.19 | 0.07 | $\mu g\ m^{-3}$ per year | -1.6 |
| **DD INVENTORY** | Number of Episodes | -0.67 | 0.03 | -1.00 | 0.00 | N.E. per year | -5.2 |
| | Number of DD event days | -2.7 | 0.02 | -4.2 | -1.30 | N.D. per year | -8.0 |
| | DD Contribution to AOD | -0.0019 | 0.016 | -0.003 | -0.000 | AOD-units per year | -11.2 |
| | DD Contribution to $PM_{10}$ | -0.14 | 0.06 | -0.26 | 0.01 | $\mu g\ m^{-3}$ per year | -10.1 |
| | DD Contribution to $PM_{2.5}$ | -0.079 | <0.01 | -0.12 | -0.04 | $\mu g\ m^{-3}$ per year | -13.7 |
| | DD Contribution to $PM_{2.5-10}$ | -0.085 | 0.06 | -0.16 | 0.00 | $\mu g\ m^{-3}$ per year | -10.0 |




Table 4. Monthly mean (and total) contribution of DD to total AOD and PMx, in absolute (AOD-units and
$\mu g\ m^{-3}$, respectively) and relative (%) values during the 2003-2013 period.

| | | Jan | Feb | Mar | Apr | May | Jun | Jul | Aug | Sep | Oct | Nov | Dec | Total |
|---|---|---|---|---|---|---|---|---|---|---|---|---|---|---|
| $AOD_{440nm}$ | abs. | 0.006 | 0.011 | 0.018 | 0.014 | 0.014 | 0.027 | 0.018 | 0.026 | 0.023 | 0.014 | 0.008 | 0.004 | 0.015 |
| | rel. | 6.05 | 9.94 | 13.38 | 9.37 | 9.86 | 17.07 | 12.07 | 17.42 | 15.66 | 12.37 | 9.55 | 5.40 | 11.51 |
| $PM_{10}$ | abs. | 0.51 | 0.58 | 2.23 | 0.81 | 0.96 | 2.28 | 2.35 | 2.38 | 1.16 | 1.37 | 0.83 | 0.23 | 1.31 |
| | rel. | 7.70 | 6.73 | 20.13 | 9.78 | 8.54 | 17.69 | 16.51 | 16.13 | 9.61 | 13.57 | 11.57 | 3.64 | 11.80 |
| $PM_{2.5}$ | abs. | 0.36 | 0.43 | 0.66 | 0.28 | 0.33 | 0.88 | 1.12 | 1.12 | 0.58 | 0.50 | 0.28 | 0.08 | 0.55 |
| | rel. | 7.85 | 7.16 | 10.50 | 5.60 | 5.00 | 11.99 | 13.64 | 13.40 | 8.33 | 9.71 | 7.26 | 1.88 | 8.53 |
| $PM_{2.5-10}$ | abs. | 0.20 | 0.17 | 1.67 | 0.47 | 0.69 | 1.40 | 1.25 | 1.33 | 0.58 | 0.93 | 0.56 | 0.17 | 0.79 |
| | rel. | 8.18 | 5.55 | 32.84 | 13.89 | 14.28 | 24.76 | 20.59 | 20.90 | 11.26 | 18.50 | 16.34 | 6.57 | 16.14 |








Table 5. Mean annual contribution of DD to total AOD and PMx in absolute (AOD-units and µg m$^{-3}$,
respectively) and relative (%) values during the 2003-2013 period.

| | | 2003 | 2004 | 2005 | 2006 | 2007 | 2008 | 2009 | 2010 | 2011 | 2012 | 2013 | 2014 |
|---|---|---|---|---|---|---|---|---|---|---|---|---|---|
| $AOD_{440nm}$ | abs. | 0.029 | 0.033 | 0.016 | 0.022 | 0.022 | 0.018 | 0.006 | 0.012 | 0.017 | 0.018 | 0.004 | 0.008 |
| | rel. | 15.87 | 21.22 | 10.76 | 15.59 | 16.06 | 14.78 | 4.68 | 11.75 | 12.03 | 14.67 | 3.76 | 7.25 |
| $PM_{10}$ | abs. | 2.29 | 2.35 | 2.07 | 2.39 | 1.18 | 1.08 | 0.50 | 0.94 | 1.11 | 1.14 | 0.39 | 1.25 |
| | rel. | 18.12 | 17.80 | 16.10 | 21.49 | 11.02 | 11.04 | 5.59 | 10.81 | 10.92 | 12.36 | 4.87 | 14.54 |
| $PM_{2.5}$ | abs. | 1.04 | 0.87 | 0.92 | 1.23 | 0.52 | 0.68 | 0.25 | 0.34 | 0.31 | 0.31 | 0.16 | 0.30 |
| | rel. | 13.07 | 10.27 | 11.95 | 17.79 | 8.10 | 10.36 | 4.76 | 6.92 | 5.99 | 6.88 | 3.52 | 6.55 |
| $PM_{2.5-10}$ | abs. | 1.38 | 1.61 | 1.21 | 1.21 | 0.73 | 0.42 | 0.27 | 0.65 | 0.71 | 0.91 | 0.18 | 1.01 |
| | rel. | 27.00 | 30.99 | 22.91 | 26.97 | 15.85 | 12.52 | 6.72 | 15.34 | 14.42 | 18.31 | 4.88 | 22.57 |




**Figure captions**
Figure 1. Location of the main sites used in this study belonging to AERONET (blue diamonds) and EMEP
(red stars) networks.
Figure 2. Annual cycle of a) total number of episodes per month for total DD intrusions; b) total (blue bars)
number of days per month for total DD intrusions and for desert (D, green bars) and mixed desert (MD, red
bars) categories. Mean values per month can be derived dividing by 12.
Figure 3. Inter-annual variability of a) total number of episodes; b) total (blue bars) number of days for the
desert dust (DD) intrusions and for desert (D, green bars) and mixed desert (MD, red bars) categories.
Figure 4. Annual cycle for DD contribution to the total monthly AOD means in absolute (red bar) and
relative values (black line). Blue bars represent the annual cycle of total AOD and green bars the
corresponding values without including the days of desert dust.
Figure 5. Inter-annual variability of DD contribution to the total yearly AOD in absolute (red bar) and
relative values (black line). Blue bars represent the mean year AOD value and green bars the corresponding
values without including the days of desert dust.
Figure 6. Annual cycle for DD contribution to the total monthly $PM_{10}$ (a), $PM_{2.5}$ (b), and $PM_{2.5-10}$ (c) means
in absolute (red bar) and relative values (black line). Blue bars represent the annual cycle of total $PM_{10}$ (a),
$PM_{2.5}$ (b), and $PM_{2.5-10}$ (c) and green bars the corresponding values without including the days of desert
dust.
Figure 7. Inter-annual variability of DD contribution to the total yearly $PM_{10}$ (a), $PM_{2.5}$ (b), and $PM_{2.5-10}$ (c)
in absolute (red bar) and relative values (black line). Blue bars represent the mean year $PM_{10}$ (a), $PM_{2.5}$ (b),
and $PM_{2.5-10}$ (c) value and green bars the corresponding values without including the days of desert dust.
Figure 8. Annual cycle (a) and inter-annual (b) variability of DD episodes classified in terms of their
synoptic scenarios: NAH-S (white bars), AD (green bars), NAD (red bars), and NAH-A (blue bars).















Figure 1

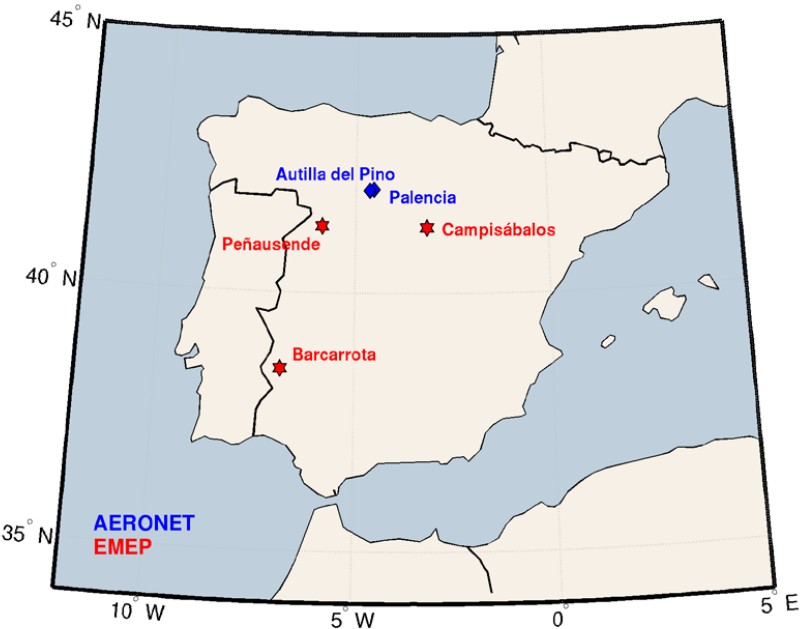


Figure 1. Location of the main sites used in this study belonging to AERONET (blue diamonds) and EMEP
(red stars) networks.






Figure 2

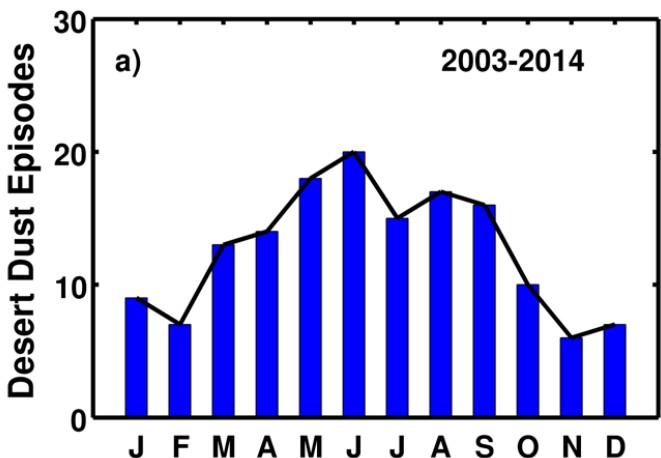


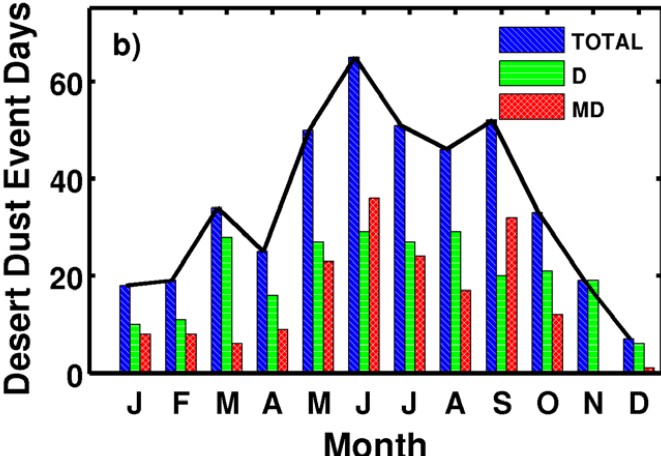


Figure 2. Annual cycle of a) total number of episodes per month for total DD intrusions; b) total (blue bars)
number of days per month for total DD intrusions and for desert (D, green bars) and mixed desert (MD, red
bars) categories. Mean values per month can be derived dividing by 12.






Figure 3

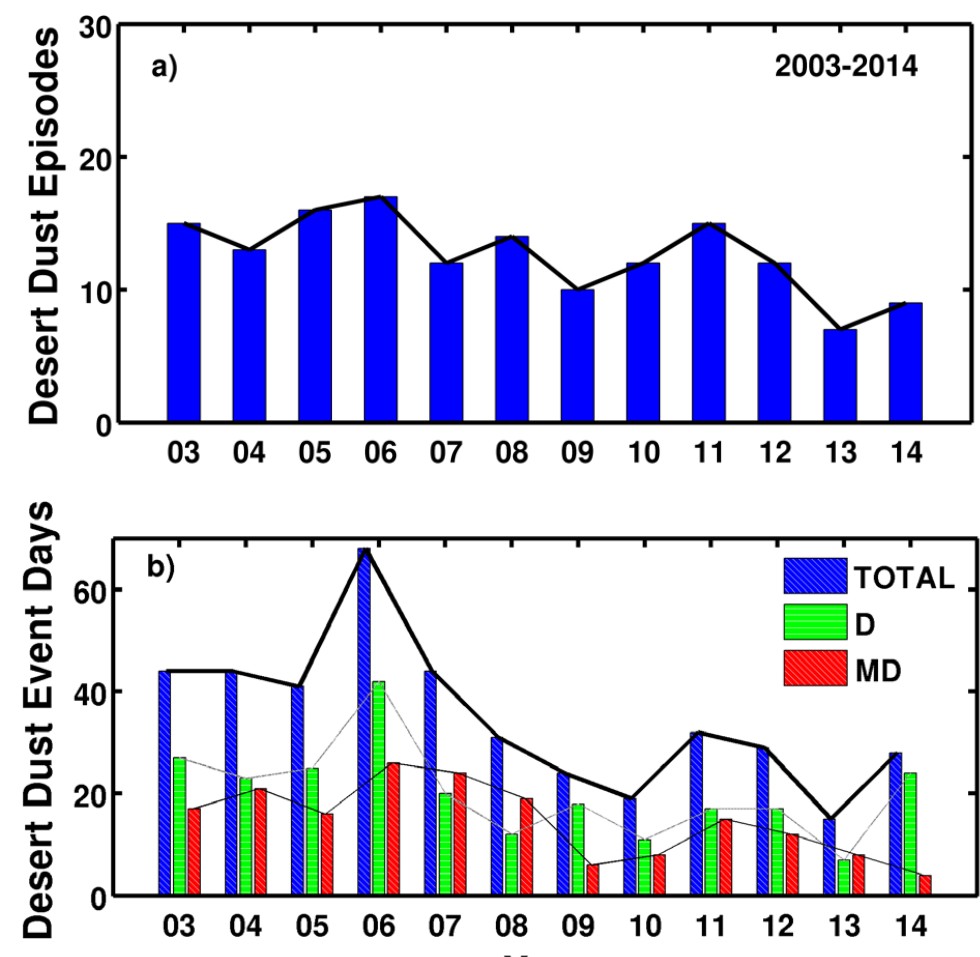


Figure 3. Inter-annual variability of a) total number of episodes; b) total (blue bars) number of days for the
desert dust (DD) intrusions and for desert (D, green bars) and mixed desert (MD, red bars) categories.



Figure 4

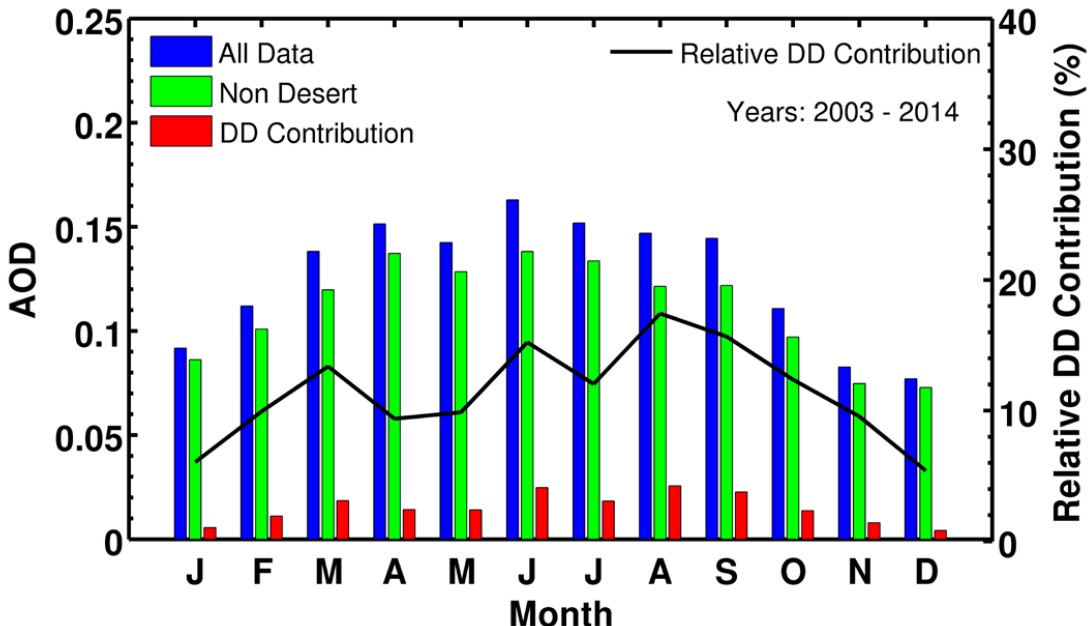


Figure 4. Annual cycle for DD contribution to the total monthly AOD means in absolute (red bar) and
relative values (black line). Blue bars (also indicated as All Data) represent the annual cycle of total AOD
and green bars the corresponding values without including the days of desert dust (indicated as Non
desert).












Figure 5

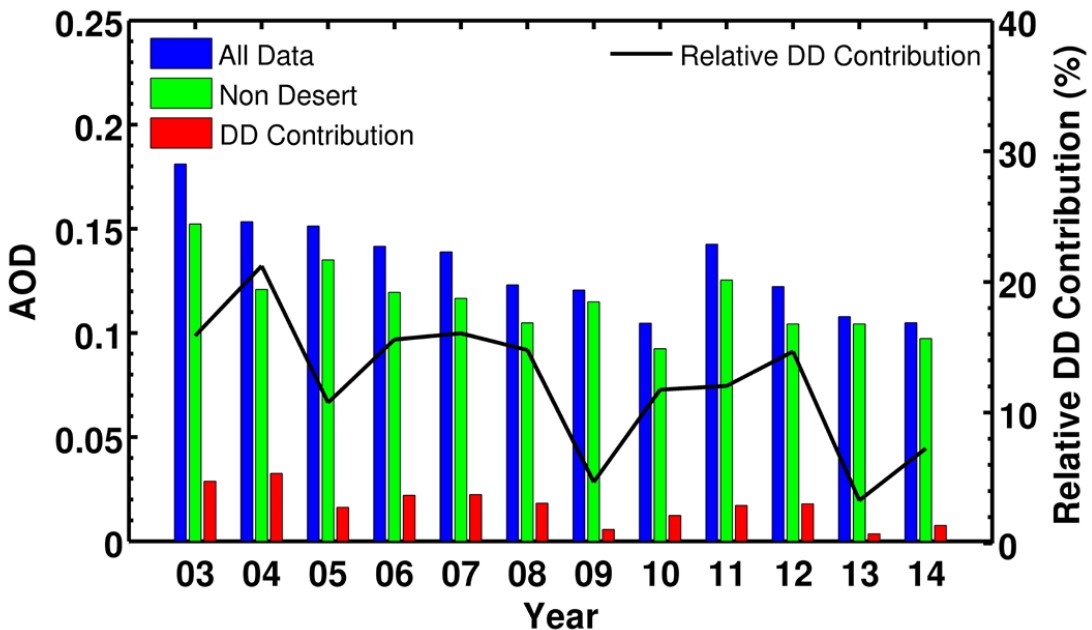



Figure 5. Inter-annual variability of DD contribution to the total yearly AOD in absolute (red bar) and
relative values (black line). Blue bars (also indicated as All Data) represent the mean year AOD value and
green bars the corresponding values without including the days of desert dust (also indicated as Non
Desert) .





Figure 6

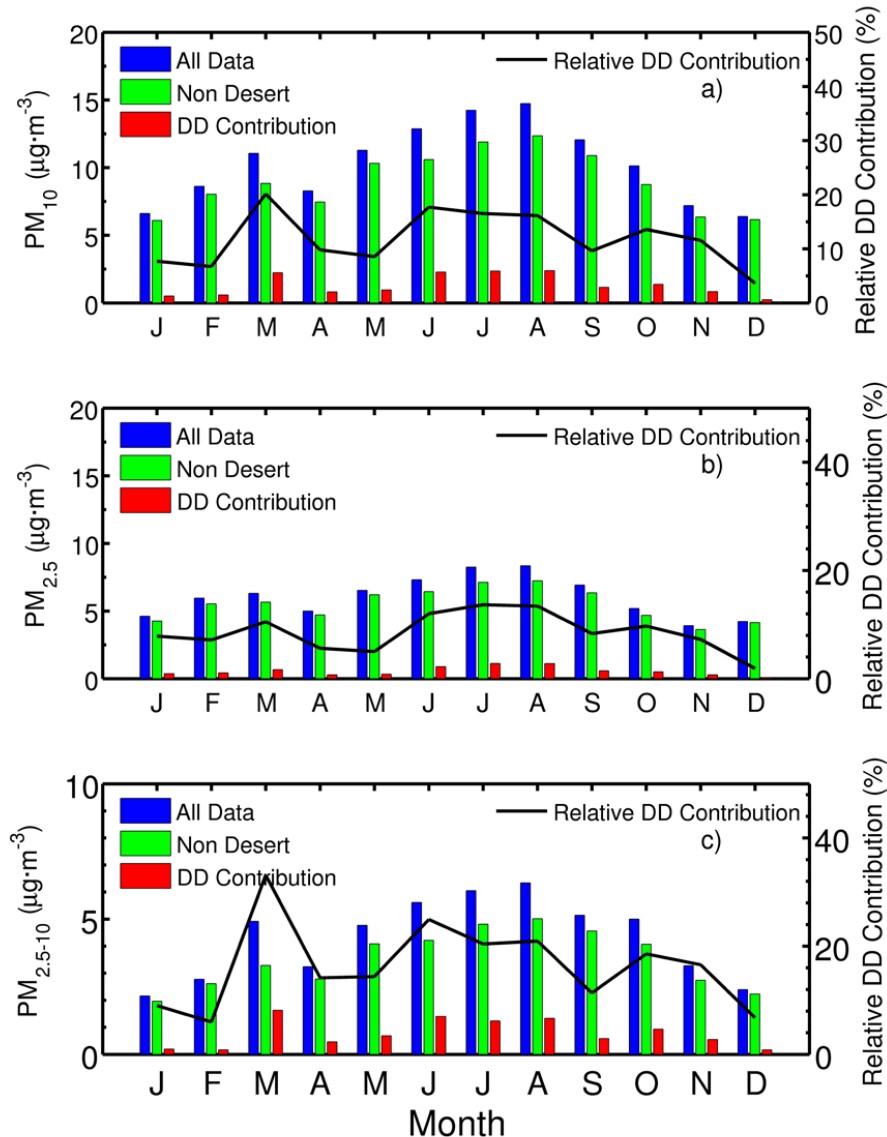


Figure 6. Annual cycle for DD contribution to the total monthly $PM_{10}$ (a), $PM_{2.5}$ (b), and $PM_{2.5-10}$ (c) means
in absolute (red bar) and relative values (black line). Blue bars represent the annual cycle of total $PM_{10}$ (a),
$PM_{2.5}$ (b), and $PM_{2.5-10}$ (c) and green bars the corresponding values without including the days of desert
dust.




Figure 7

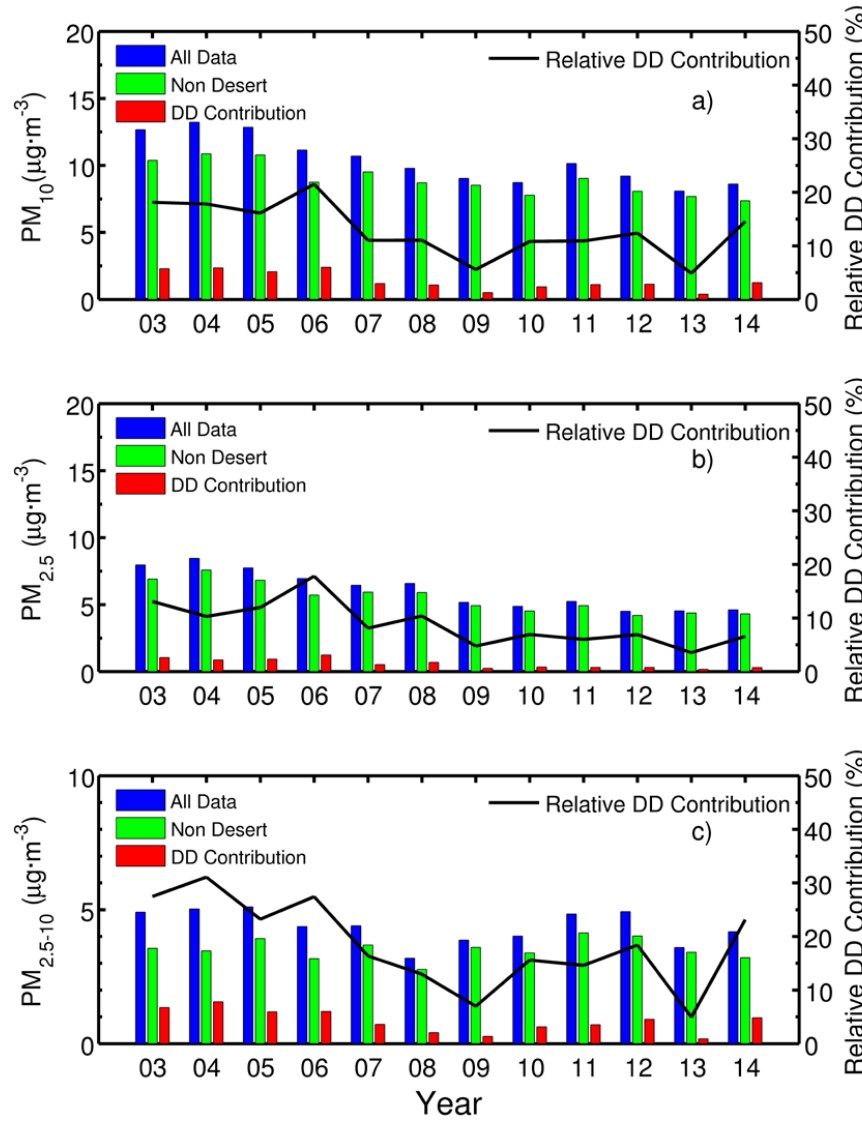


Figure 7. Inter-annual variability of DD contribution to the total yearly PM$_{10}$ (a), PM$_{2.5}$ (b), and PM$_{2.5\text{-}10}$ (c)
in absolute (red bar) and relative values (black line). Blue bars represent the mean year PM$_{10}$ (a), PM$_{2.5}$ (b),
and PM$_{2.5\text{-}10}$ (c) value and green bars the corresponding values without including the days of desert dust.




Figure 8


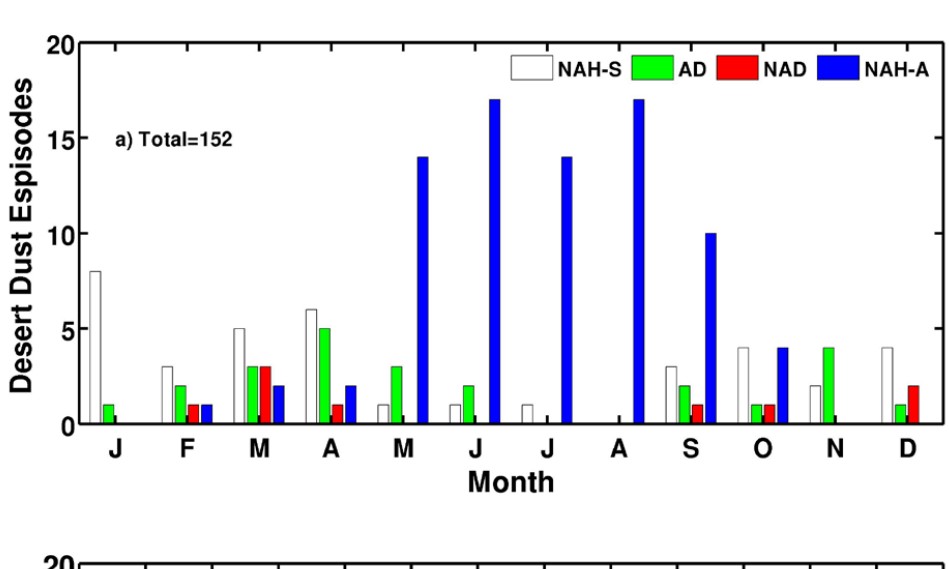


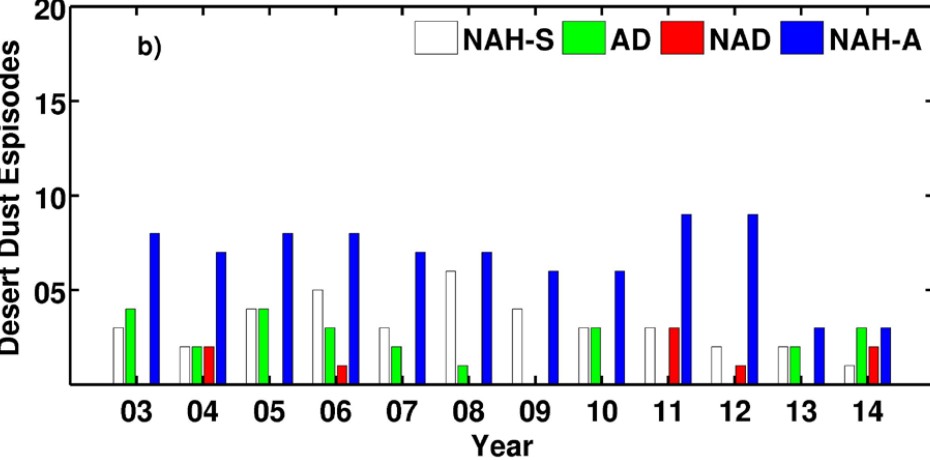


Figure 8. Annual cycle (a) and inter-annual (b) variability of DD episodes classified in terms of their
synoptic scenarios: NAH-S (white bars), AD (green bars), NAD (red bars), and NAH-A (blue bars).