# Peer review of "Inventory of African desert dust events in the north-central Iberian Peninsula in 2003-2014 based on Sun photometer-AERONET and particulate mass-EMEP data"

_Atmospheric Chemistry and Physics, 2016_

## Referee Comment (RC1) · Anonymous Referee #1 · 22 Mar 2016

This paper is very comprehensive and complete, even too lengthy at some points. It is also well written and I recommend it for publication as is.

The only note is to check the name of O'Neill through the paper.

---

## Referee Comment (RC2) · Anonymous Referee #2 · 26 Apr 2016

The paper "Inventory of African desert dust events in the North-central Iberian peninsula in 2003-2014 based on sunphotometer-aeronet and particulate mass-emet data" by Cachorro et al., is an interesting paper dealing with a very timely and interesting topic: the dust presence over Southern Europe and its impact on PM measurements. There are some aspects however of the data analysis and discussion that should be improved for providing the readers with more robust results. My main comment is about the first criterion for selecting dust cases: a threshold on AOD. Firstly, the mean value of

AOD is a mean for dust+ no dust cases, since cases including dust cases are within the dataset used for calculating the mean. Therefore the threshold mean + st dev selects by definition only extreme values, that at this stage could be dust or no dust cases. The same concept applies to the threshold on Angstrom exponent. There are many studies showing the occurrence of low dust AOD cases over Europe (example at network level Papayannis et al., 2008). Of course threshold criteria are always questionable but probably authors could improve the results considering seasonal threshold. About the threshold issue, the big question here is: the authors found a decreasing in the aod , i.e. this is a no stationary condition for this aerosol parameter and the threshold cannot be define in this way. The result here presented about the decreasing of dust cases is simply the consequence of the fixed threshold defined. The authors should consider the decreasing in AOD in the threshold application. Without taking into account the AOD trends results of the study would be compromised. My suggestion is defining a season by season threshold and reanalyzing data in view of this modification.

Minor comments are reported in the following. Line 134: "PMx sampling is typically ... " there are many cases in which the PMx is provided with high temporal resolution

Line 151: other papers combining columnar AOD and PMx measurements are available in literature, reviewing this paper I found very interesting the paper Boselli et al., Atm Env. which shows how cluster analysis in the aod/Angstrom space could be good for the dust identification with certain limit when this is compared to air mass analysis showing also that dust cases are very variable in AOD/Angstrom values.

Line 155: probably reliable is not the correct word here.

Line 222: Complete is here too much, sufficient, satisfying?

Line 233: are 5 day backtrajectories sufficient? Iberian peninsula is not so far away from African, but however there could be longer paths carrying dust from the desert.

Line 279-281: include references for these values

Figure 4: this could be compared to AOD related just to dust layers as obtained by lidar observations available in literature.

---

## Author Comment (AC2) · 20 May 2016

We are herewith providing our replies with suitable answers to the comments raised by the reviewer #2. Our replies are highlighted in red colour font. At the end of the reply, the reviewer can find the new version of the manuscript with all the changes marked.

**Anonymous Referee #2**

The paper "Inventory of African desert dust events in the North-central Iberian peninsula in 2003-2014 based on sunphotometer-aeronet and particulate mass-emet data" by Cachorro et al., is an interesting paper dealing with a very timely and interesting topic: the dust presence over Southern Europe and its impact on PM measurements. There are some aspects however of the data analysis and discussion that should be improved for providing the readers with more robust results. My main comment is about the first criterion for selecting dust cases: a threshold on AOD. Firstly, the mean value of AOD is a mean for dust+ no dust cases, since cases including dust cases are within the dataset used for calculating the mean. Therefore the threshold mean + st dev selects by definition only extreme values, that at this stage could be dust or no dust cases. The same concept applies to the threshold on Angstrom exponent. There are many studies showing the occurrence of low dust AOD cases over Europe (example at network level Papayannis et al., 2008). Of course threshold criteria are always questionable but probably authors could improve the results considering seasonal threshold.

We have re-written the description of the methodology in order to avoid confusions or misleading.  As the reviewer can see in the new description (see below), our method does not need a seasonal threshold since the visual inspection by human observers is daily performed. The established thresholds are assumed as "warning values" in order to more carefully analyze those days with possible presence of desert dust. Therefore, the final decision of considering a dusty day is carried out by a human observer taking into account all the information explained in the manuscript (AOD, PM10, air mass back-trajectories, satellite images, model forecast). This comprehensive method is difficult to apply in wider areas with many sites but this inventory can be considered as a reference for validation purposes.

So, our results are stable and they are not seasonal dependent.  As we have mentioned in the new version of the manuscript, the thresholds are considered as "warning flags" and never are used alone in the DD identification. We have also identified desert dust intrusions showing a very low intensity with daily means of AOD and $PM_{10}$ notably smaller than the corresponding thresholds (see Figures below). All the reported temporal trends are consistent and are attributed to a real decrease in the number of DD episodes as well as their contribution to aerosol load in the 2003-2014 period.

The text added is the following:

"Certain thresholds have been established in order to identify those conditions which are separated from the clean background over the study area. Hence, these choices are based on the aerosol climatology of the site from our investigations (Bennouna et al., 2014; Mateos et al., 2015) and previous results (see, e.g., Querol et al., 2009; 2014). The mean values for the long-term period 2003-2014 are 0.13 ± 0.09 for $AOD_{440nm}$ and 10±9 µg m$^{-3}$ for $PM_{10}$. In order to

detect the DD intrusions over the study area, a visual inspection of the entire database is performed. When a day shows a group of the instantaneous AOD ≥ 0.18 and/or the daily $PM_{10}$ ≥ 13 µg m$^{-3}$, that day is carefully investigated. Hence, these thresholds must be taken as "warning flags" in the sense that these values alone do not define the classification as a dusty day. They also need the ancillary information given by the air mass backwards trajectories, satellite images, weather maps, and model forecasts in order to determine and corroborate the origin of aerosols and synoptic conditions. Therefore, with all this information the human observer decides if this day must be included or not in the DD inventory.

Simultaneously to the former analysis, the evolution of α quantity is also checked, allowing the identification of two different types of DD intrusions. Those days displaying α ≤ 1.0 in most of the instantaneous columnar data are identified as the "purer" desert dust intrusions and they are denoted by D flag. Those days with α values in the interval 1.0 ≤ α ≤ 1.5 (with the ensured presence of desert dust aerosols by the ancillary information) present a mixture with other types and they are denoted by MD (Mixed Desert) flag. The MD event days may either be a part of an intense event (generally of D type) or form by themselves a low-moderate intensity event. Previous studies have also stated that DD intrusions in the Mediterranean Basin can present moderate AOD associated with large α values (e.g., Pace et al., 2006; Tafuro et al., 2006; Boselli et al., 2012). The limit of α ≤ 1 for identifying coarse particles has been established by previous studies in different areas (e.g., Eck et al., 1999; 2010; Dubovik et al., 2002, Meloni et al., 2007; Boselli et al., 2012), making this threshold suitable for our study area. It is important to emphasize here that the α parameter allows a more accurate identification of desert dust events, mainly those of low intensity with a less desert character (overall, the larger the AOD of a desert event, the lower the α) generally mixed with other aerosol types which are not accounted for in many DD studies (e.g., Gkikas et al., 2013). The DD events of low intensity are more difficult to detect and hence it is more difficult to evaluate their contribution or impact on AOD and PMx daily values. In fact, the aerosol load threshold used in this study could seem very low, but we must note again here that all these quantities are manually supervised by the expert-observer who will take the final decision of the inclusion or not of a dusty day, having in mind all the available information given by these data and all the complementary material."

About the threshold issue, the big question here is: the authors found a decreasing in the aod , i.e. this is a no stationary condition for this aerosol parameter and the threshold cannot be define in this way. The result here presented about the decreasing of dust cases is simply the consequence of the fixed threshold defined. The authors should consider the decreasing in AOD in the threshold application. Without taking into account the AOD trends results of the study would be compromised. My suggestion is defining a season by season threshold and reanalyzing data in view of this modification.

During the DD identification, the human observer bears in mind the seasonal evolution. A new paper dealing with the aerosol optical and microphysical characterization of DD events has been recently sent for publication as a part of this comprehensive DD aerosol study. In that paper we analyzed the intensity of the DD episodes in the inventory, the reviewer can see this quantity in the following α vs AOD plot and the frequency histograms of AOD and PM10 values:

[Figure]

As it is mentioned in the text and it is shown in these figures, we have identified desert dust events with low intensity (daily values notably lower than the corresponding threshold). The use of both columnar and surface data provides a better indicator of the DD intrusions over our study area. Our method, of course, presents a certain uncertainty as all the DD inventories, but we present in the manuscript a reasonable evaluation of it.

As it is mentioned above, the trends in both dusty days and DD contribution are consistent and there are less DD event days which also show lower intensity at the ending of the analyzed period.

Minor comments are reported in the following.

Line 134: "PMx sampling is typically ... " there are many cases in which the PMx is provided with high temporal resolution

The reviewer is right, we've added: "The PMx sampling used here is based on daily records…"

Line 151: other papers combining columnar AOD and PMx measurements are available in literature, reviewing this paper I found very interesting the paper Boselli et al., Atm Env. which shows how cluster analysis in the aod/Angstrom space could be good for the dust identification with certain limit when this is compared to air mass analysis showing also that dust cases are very variable in AOD/Angstrom values.

We have added the mentioned reference since that study also reinforces the occurrence of mineral dust with low intensities and large Angstrom values.

Line 155: probably reliable is not the correct word here.

We have deleted 'reliable'

Line 222: Complete is here too much, sufficient, satisfying?

This sentence has been changed to: "The use of AOD440nm, α, PM10, and PM2.5 observations provides a comprehensive database…"

Line 233: are 5 day backtrajectories sufficient? Iberian peninsula is not so far away from African, but however there could be longer paths carrying dust from the desert.

Following the study by Pace et al. (2006), the use of 3-day or 7-day trajectories does not significantly modify the average optical properties of each class with respect to the 5-day trajectories. Hence, we have assumed that 5-days can be an adequate time scale for the origin of air masses.

Line 279-281: include references for these values

Done. We have included references in the reviewed version.

Figure 4: this could be compared to AOD related just to dust layers as obtained by lidar observations available in literature

It is difficult to compare the AOD related to dust layers with the long-term DD contribution presented in Figure 4 since the quantities are not the same. The former gives an idea about the intensity, which is given in most of the case studies analyzed in literature. We have added previous studies using LIDAR database in the discussion of the annual cycles of dusty days in certain areas of the Iberian Peninsula and also in the Mediterranean Basin. Unfortunately, any LIDAR is available in the study area to perform a reliable comparison with the Sun photometer data. The closest LIDAR is placed in Madrid city, which could not be representative of the entire area because of the Central System landform. As we mentioned in the text (see Section 4.1.1), previous studies of DD inventories using surface aerosols in Madrid area have obtained a larger occurrence of DD outbreaks than the presented in our study.

[revised manuscript text omitted]